# Genome-wide association study and polygenic risk prediction of hypothyroidism

Søren A. Rand [1,2] ✉, Gustav Ahlberg [2,3], Vinicius Tragante [4], Laia M. Monfort [2,3], Chaoqun Zheng [1], Ulla Feldt-Rasmussen[5,6], Marianne C. Klose[6], Maris Teder-Laving [7], Andres Metspalu [7], Henrik E. Poulsen [5,8,9], Christina Ellervik [5,10,11], Birte Nygaard [5,12], Christian Erikstrup [13,14], Mie T. Bruun [15], Bitten A. Jensen[16], Henrik Ullum[17], Søren Brunak[18,19], DBDS Genomic Consortium*, Estonian Biobank Research Team*, 23andMe Research Team*, Michael Schwinn [20], Sisse R. Ostrowski [5,20], Ole B. Pedersen [5,21], Erik Sørensen [20], Ingileif Jonsdottir [4,22], Daniel F. Gudbjartsson [4], Gudmar Thorleifsson [4], Hilma Holm [4], Saedis Saevarsdottir [4,22], Kari Stefansson[4,22], Morten Salling Olesen [2,3], Henning Bundgaard [1,5,31] & Jonas Ghouse [1,14,23,24,31] ✉

We performed a genome-wide meta-analysis of hypothyroidism (113,393 cases and 1,065,268 controls), free thyroxine (191,449 individuals) and thyroid-stimulating hormone (482,873 individuals). We identified 350 loci associated with hypothyroidism, including 179 not previously reported, 29 of which were linked through thyroid-stimulating hormone. We found that many hypothyroidism risk loci regulate blood cell counts and the circulating inflammasome, and through multiple gene-mapping strategies, we prioritized 259 putative causal genes enriched in immune-related functions. We developed a polygenic risk score (PRS) based on more than 115,000 hypothyroidism cases to address diagnostic challenges in individuals with or at risk of thyroid hormone deficiency. We show that the highest predictive accuracy for hypothyroidism was achieved when combining the PRS with thyroid hormones and thyroid-peroxidase autoantibodies, and that the PRS was able to stratify risk of progression among individuals with subclinical hypothyroidism. These findings demonstrate the potential for a hypothyroidism PRS to support the prediction of disease progression and onset in thyroid hormone deficiency.

Primary hypothyroidism is a common and insidious metabolic disease. It is characterized by subtle and nonspecific symptoms, which can lead to delayed diagnosis, resulting in an underdiagnosed case burden estimated at up to 0.5%[1,2]. Thyroid hormone deficiency is associated with increased risk of cardiometabolic outcomes, including coronary artery disease (CAD), heart failure (HF) and metabolic syndrome[1,3,4]. The risk of hypothyroidism is influenced by various factors, such as subclinical hypothyroidism (SCH), autoimmunity,

iodine supplementation, sex, age, ancestry and goiter[1,5,6]. Genetics play an important role, with twin studies estimating that 55% of the disease risk may be attributed to genetic factors[7], and genome-wide association studies (GWAS) have linked hundreds of genetic loci to thyroid disease and related biomarkers[8–14].

Screening for thyroid dysfunction is standard in clinical practice, with up to 25% of some populations undergoing annual thyroid function tests[5]. The diagnosis of overt hypothyroidism is

**Fig. 1 | Hypothyroidism lead variants and their associations with thyroid hormones. a**, Relationships between minor allele frequencies and ORs for the 350 lead variants that were identified in the hypothyroidism genome-wide meta-analysis (113,393 cases and 1,065,268 controls) or through an endophenotype-driven analysis using thyroid-stimulating hormone genome-wide associations as priors. Coding variants are squared, new associations are turquoise, and known associations are gray. **b**, Relationships between hypothyroidism risk and changes in thyroid-stimulating hormone for 349 of 350 lead variants. **c**, Relation between hypothyroidism risk and change in free thyroxine for 348 of 350 lead variants. In **b** and **c**, the centerline represents the linear regression, and the shaded error band shows the 95% CI around the regression line. Statistical associations were assessed using two-sided Pearson correlation tests. No multiple testing correction was applied for these correlation analyses.

straightforward. However, individuals with SCH characterized by elevated thyroid-stimulating hormone (TSH, >4 mU l$^{-1}$) and free thyroxine (fT4) within the reference range pose a diagnostic challenge. Current guidelines recommend treating SCH with thyroid hormone replacement when TSH exceeds 10 mU l$^{-1}$, if the patient is young, have a positive screen for thyroid-peroxidase antibodies (anti-TPO), have cardiovascular disease or exhibit symptoms of hypothyroidism[4]. Notably, relying solely on symptoms for treatment decisions may lead to overdiagnosis and overtreatment since classic hypothyroidism manifestations (for example, lethargy, dry skin or impaired memory) are commonly observed in euthyroid individuals[4]. Similarly, basing treatment decisions solely on biochemical findings may result in overtreatment since more than one-third of patients with abnormal thyroid function tests experience spontaneous remission without intervention[15]. In addition to biochemical testing, no risk assessment tool can distinguish between patients with high and low risk of disease progression. Given the high heritability and polygenic nature of hypothyroidism, we hypothesized that a well-powered polygenic risk score (PRS), incorporating millions of sequence variants, could aid in identifying high-risk individuals.

This GWAS meta-analysis, which included 113,393 hypothyroidism cases, 1,065,268 controls and over 190,000 individuals with measured thyroid hormone levels, offers insights into the genetic underpinnings of thyroid hormone deficiency. We characterized the hypothyroidism immunophenotype by investigating genetic associations with peripheral blood cell counts and circulating levels of inflammatory markers.

We developed a PRS to improve the prediction of hypothyroidism and compared the predictive ability in incident disease relative to traditional risk factors. We then evaluated the ability of the PRS to predict progression from subclinical to overt hypothyroidism. Finally, we explored the association between the hypothyroidism PRS and common malignancies, cardiometabolic and neuropsychiatric traits.

## Results

### Genome-wide association results

We included 113,393 hypothyroidism cases and 1,065,268 controls from European cohorts in the GWAS meta-analysis of hypothyroidism (Copenhagen Hospital Biobank-Chronic Inflammatory Diseases/Danish Blood Donor Study (CHB-CID/DBDS), UK Biobank (UKB), FinnGen and 23andMe). The genomic inflation factor ($\lambda_{GC}$) was 1.46, and the linkage-disequilibrium score regression (LDSC)-intercept was 1.09 (s.e. = 0.03; Supplementary Table 1), indicating that most of the observed inflation was due to polygenicity. At genome-wide significance ($P < 5 \times 10^{-8}$), we identified 319 loci, of which 150 were previously unreported (Fig. 1a and Supplementary Tables 2 and 3) and 84 were not previously associated with other thyroid traits (Supplementary Table 4). Using a more stringent threshold of $P < 1 \times 10^{-9}$, we found 247 loci, of which 86 were unreported. The heritability was 14.5% (95% confidence interval (CI) = 14.0–15.0). Most lead single-nucleotide polymorphisms (SNPs) had modest effect sizes (median odds ratio (OR) = 1.03, interquartile range = 0.96–1.05). We discovered several low-frequency

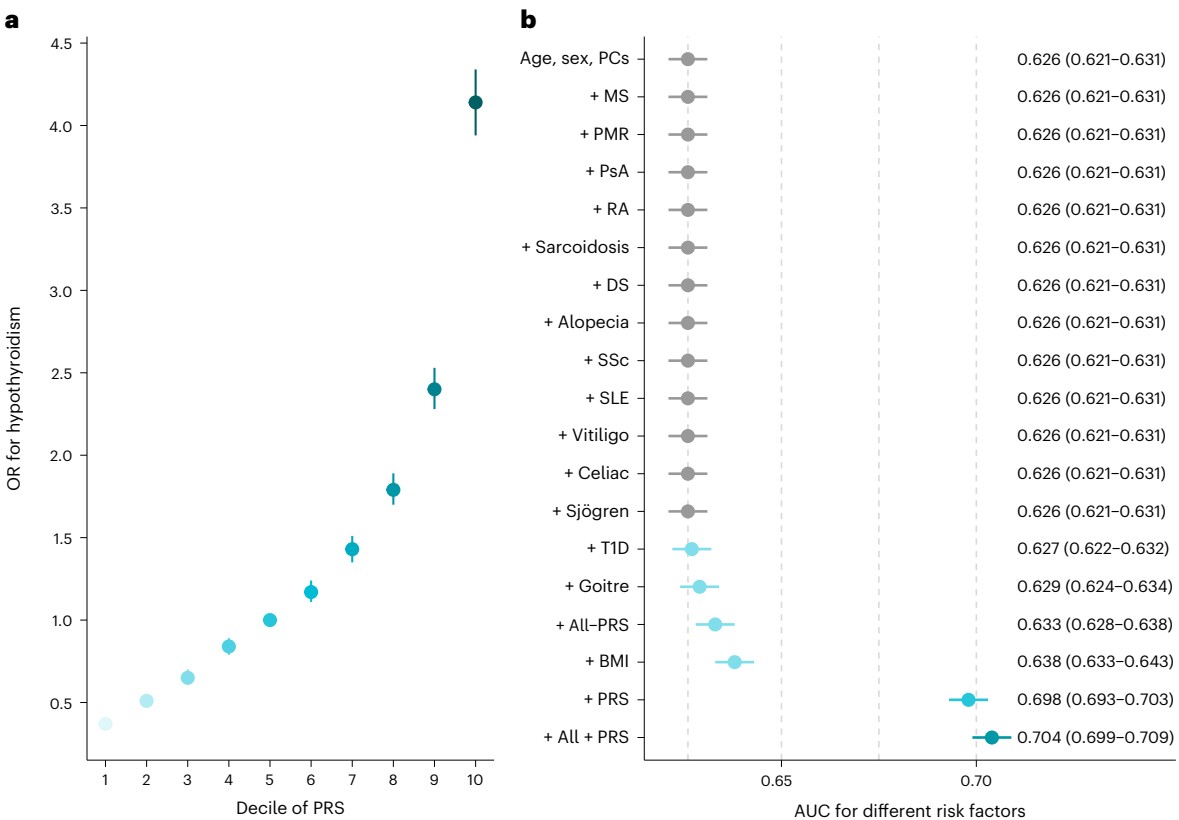

**Fig. 2 | PRS association with and prediction of hypothyroidism.**
**a**, Associations between 10 deciles of the PRS and risk of hypothyroidism are presented as OR point estimates ± 95% CI error bars, estimated using logistic regression models adjusted for age, sex and PCs. No adjustments were made for multiple comparisons. **b**, Prediction of incident hypothyroidism cases. The benchmark model consisted of age, sex and four PCs. Prevalent risk factors for hypothyroidism were added iteratively to the benchmark model.

The center of each error bar represents the AUC, and error bars indicate the 95% CIs, displayed in absolute terms on the right. No adjustments were made for multiple comparisons. MS, multiple sclerosis; PMR, polymyalgia rheumatica; PsA, psoriatic arthritis; RA, rheumatoid arthritis; DS, Down syndrome; SSc, systemic sclerosis; SLE, systemic lupus erythematosus; Celiac, celiac disease; Sjögren, Sjögren's disease; T1D, type 1 diabetes.

(minor allele frequency (MAF) < 5%) coding variants, which included a known stop-gain in *TSHR* (p.Trp546Ter; OR = 7.67, rs121908866) and two new and protective missense variants—rs149007883 in *NFKBIZ* (p.Gly102Ala; OR = 0.83, $P = 4.94 \times 10^{-8}$) and rs61731111 in *S1PR4* (p.Arg243Cys; OR = 0.91, $P = 8.66 \times 10^{-9}$).

### Endophenotype-driven analysis

We meta-analyzed GWASs of thyroid hormones from CHB-CID/DBDS, UKB and previously published summary data[10,16]. In a meta-analysis of up to 191,449 individuals with fT4 measurements, we identified 61 fT4 genome-wide significant loci, of which 15 were previously unreported (Supplementary Table 5). In a meta-analysis of up to 482,873 individuals with TSH measurements, 297 TSH genome-wide significant loci were identified, 126 of which have not been previously reported (Supplementary Table 6). Using LD score regression, we found that the genetic correlations with hypothyroidism were 55% ($P = 3.55 \times 10^{-122}$) for TSH and −23% ($P = 3.95 \times 10^{-3}$) for fT4. Based on the strong link between TSH and hypothyroidism, we used TSH GWAS associations as priors to enhance genomic discovery for hypothyroidism. Of the 297 TSH variants, 186 were associated with hypothyroidism at a false discovery rate (FDR) < 0.01. Of these, 96 were previously associated with hypothyroidism at genome-wide significance, 61 overlapped in positions with genome-wide hypothyroidism loci reported in this study and the remaining 29 represent new associations for hypothyroidism. In total, we identified 350 nonoverlapping loci via hypothyroidism meta-analysis or through the TSH-driven approach (Supplementary Table 7), 179 of which have not been reported previously.

### Replication

We replicated unreported variants in the Estonian Biobank (EstBB) and deCODE genetics, which included 34,835 cases and 492,149 controls. Of the 179 new loci reported here, 176 (98%) were available for replication. In total, 35 of 176 (19%) variants replicated beyond the threshold for multiple testing ($P < 2.79 \times 10^{-4}$ (0.05/179)). A total of 110 of 176 (63%) were nominally significant ($P < 0.05$), and all but one had concordant direction of effect. Finally, 54/176 (31%) had $P \geq 0.05$ but showed concordant direction of effect. There was a high concordance between effect estimates in the discovery and replication cohorts for the 179 risk variants ($r^2 = 0.85$, $P = 6.54 \times 10^{-51}$). Given the large sample size differences between discovery and replication, we did not expect to be able to replicate all new loci at the threshold for multiple testing. Power calculations indicated that our replication analysis had limited power to detect variants with an OR of 1.03, which corresponds to the effect range of replication variants (Supplementary Fig. 1 and Supplementary Table 8). We also cross-referenced variants that replicated at nominal significance ($P < 0.05$) with genome-wide associations to TSH and fT4. Of the 75 variants that replicated at nominal significance ($P < 0.05$), 32 were previously genome-wide significant in either TSH or fT4 studies. Of the 54 variants that did not replicate ($P \geq 0.05$) but had concordant direction of effect, 23 were genome-wide associated with either TSH or fT4 (Supplementary Table 9).

### Correlation between hypothyroidism and thyroid hormones

Since the diagnosis of hypothyroidism is biochemical, we investigated the influence of hypothyroidism variants on thyroid hormone levels.

We observed a modest correlation between the effect of hypothyroidism variants and TSH effect estimates (Pearson's $r = 0.58$, $P = 3.65 \times 10^{-33}$), where 91% (315/348) of variants had concordant direction of effect. However, some notable differences existed. For example, the missense variant rs78534766 in *ADCY7* and the *FLT3* variant rs76428106 associated with large effects on hypothyroidism (OR = 1.4 and 1.37, respectively) but had a comparably small effect on TSH levels (s.d. = 0.04 and 0.08, respectively). Similarly, the variants rs2016105 in *ELK3* ($\beta = 0.17$ s.d.) and rs6885099 in *PDE8B* ($\beta = 0.16$ s.d.) had large effects on TSH but associated with a modest increase in disease risk (OR ~ 1.1; Fig. 1b and Supplementary Table 10). For fT4 levels, we found a weak correlation between disease risk and fT4 levels (Pearson's $r = -0.16$, $P = 0.004$; Fig. 1c).

### Inflammatory traits associated with hypothyroidism variants

To investigate the role of hypothyroidism variants in autoimmunity, we tested associations between lead variants, peripheral blood cell counts (for example, red blood cells, platelets, lymphocytes, eosinophils and neutrophils) and 90 inflammatory proteins[17]. We found that 153 of the 350 (44%) lead variants were associated with blood cell traits, and 55 of the 343 lead variants that were available in protein quantitative trait locus (pQTL) data were associated with at least one inflammatory marker at $P < 1.41 \times 10^{-6}$ (0.05/350 x 101 traits; Supplementary Tables 11 and 12). The inflammatory markers with the highest number of associations with hypothyroidism lead variants were IL12B (14/55), and FLT3LG (9/55), in line with previous findings[8]. We found 40 variants associated with both blood cell traits and inflammatory proteins, with evidence of *trans*-pQTL hotspots at several loci. The lead variants with the highest number of associations were the known missense variant rs3184504 (OR = 1.18, p.Trp262Arg) in *SH2B3* and the intron variant rs11066320 (OR = 1.14) in *PTPN11*, which both associated with higher blood cell counts, and at least 30 different markers of inflammation (including several chemokines, interleukins and cytokines)[17]. Next, we interrogated variants associated with lower hypothyroidism risk in genes with known roles in immune system function. We highlight two missense variants, rs149007883 in *NFKBIZ* (p.Gly102Ala; OR = 0.83) and rs34536443 (p.Pro1104Ala; OR = 0.87) in *TYK2*, and two intron variants rs13181561 (OR = 0.96) in *STING1* and rs113473633 (OR = 0.90) in *NKFB1*. These variants were associated with lower levels of a panel of inflammatory mediators (Supplementary Fig. 2), including IFN-γ, CXCL10 and CXCL9, which make up key pathogenic pathways involved in autoimmune diseases related to hypothyroidism[18,19].

### Gene mapping

We used five methods (polygenic priority score (PoPS), variant-to-gene (V2G), coding variants, transcriptome-wide association study (TWAS) with colocalization and Mendelian disease enrichment) using different strategies (coding impact, regulatory potential and gene–trait linkage) to prioritize genes. We found 88 coding variants in 59 genes that were either lead variants (11/88) or in high LD ($r^2 > 0.8$) with one (Supplementary Table 13). Using PoPS, we mapped 209 of 350 (60%) hypothyroidism loci to 418 genes with a PoPS score in the >90th distribution, while V2G mapped 344 of 350 (98%) lead variants to a single gene (Supplementary Tables 14 and 15). Using TWAS with colocalization, we identified 272 genes within 135 of 350 (39%) hypothyroidism risk loci that showed evidence of colocalization between gene expression and disease risk (Supplementary Table 16). We found that 168 of 350 (48%) hypothyroidism loci overlapped in positions with 278 Mendelian disease genes implicated in autoimmunity or thyroid disease. Finally, 205 hypothyroidism loci had at least two lines of gene-mapping evidence prioritizing 259 genes (Supplementary Table 17). Gene enrichment analysis highlighted several genes involved in pathways related to a broad range of functions in the immune system (for example, differentiation, activation

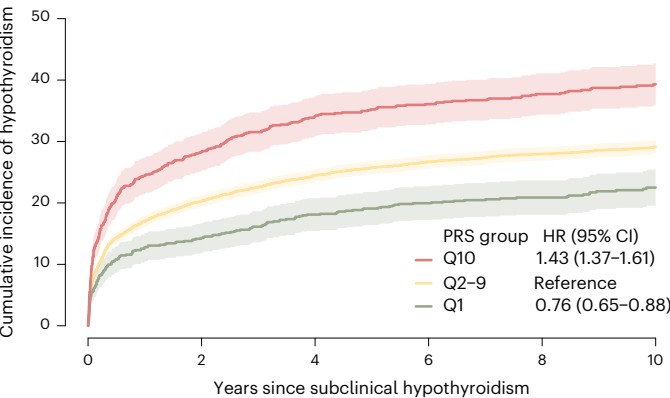

**Fig. 3 | Progression from SCH to overt disease.** Ten-year cumulative incidence of disease progression from SCH to overt hypothyroidism in 8,114 primary care patients from the UKB. Lines represent the cumulative incidence, and shaded bands indicate the 95% CI. The green line represents individuals with low polygenic risk (<10th percentile), yellow represents intermediate polygenic risk (10th–90th percentile), and red represents high polygenic risk (>90th percentile). Cumulative incidence was estimated using the Aalen-Johansen estimator, which accounts for the competing risk of death. HRs with 95% CIs were estimated using two-sided Cox proportional hazards models, adjusted for age, sex and four PCs. No adjustments were made for multiple comparisons.

and regulation of myeloid and lymphoid blood cells, regulation of cell-cell adhesion, regulation of inflammatory responses and cytokine signaling; Supplementary Table 18), but only a handful genes were enriched in thyroid hormone metabolism (for example, *GATA3*, *TPO*, *DIO1* and *TG*) or thyroid gland development (for example, *FGF10*, *TG*, *NKX2-1* and *THRA*).

### Converging effects of common and rare variants

Identifying rare coding variants in genes linked to hypothyroidism can confirm putative causal genes and increase the understanding of disease mechanisms. Using a published rare variant burden analysis including 18,362 cases and 310,690 controls[20], we investigated the associations of genes with at least two lines of mapping evidence (259 genes) and hypothyroidism, using both predicted loss-of-function variants (pLoF) and protein-altering variants (PAVs; that is, deleterious missense variants and pLoF) at an FDR-adjusted $P < 0.05$ (Supplementary Table 19). pLoF variants in *TSHR*, an established monogenic cause of hypothyroidism, were associated with increased disease risk (MAF < 1%−OR = 3.02, 95% CI = 2.25–4.06, $P = 2.5 \times 10^{-13}$). In comparison, pLoF variants in *NFATC1* (MAF < 0.001%−OR = 4.36, 95% CI = 2.11–8.99, $P = 6.7 \times 10^{-5}$) were associated with higher effect compared to *TSHR* pLoF variants, suggesting a potential monogenic role in hypothyroidism. Protective rare coding variants are particularly interesting, as they proxy effects of life-long therapeutic inhibition and may guide therapeutic developments[21]. Coding variants in four genes prioritized from our gene-mapping strategy associated with reduced risk of hypothyroidism−*ZAP70* (PAVs, MAF < 0.001%−OR = 0.33, 95% CI = 0.19−0.57, $P = 6.4 \times 10^{-5}$), *ARHGAP9* (PAVs, MAF < 1%−OR = 0.76, 95% CI = 0.66−0.87, $P = 1.3 \times 10^{-4}$), *TYK2* (PAVs, MAF < 1%−OR = 0.78, 95% CI = 0.69−0.88, $P = 6.9 \times 10^{-5}$) and *IFIH1* (pLoF, MAF < 1%−OR = 0.81, 95% CI = 0.75−0.86, $P = 2.2 \times 10^{-6}$).

### PRS and hypothyroidism prediction

We derived a PRS of 1,107,248 variants from a meta-analysis of CHB-CID/DBDS, deCODE genetics, EstBB, FinnGen and 23andMe, including more than 116,000 hypothyroidism cases. The PRS was evaluated in the UKB and the Danish General Suburban Population Study (GESUS; Methods). In the UKB, which includes 32,304 cases and 399,000 controls, we found a strong association between the PRS and

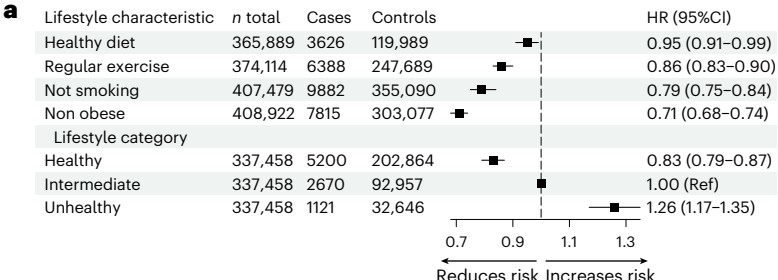

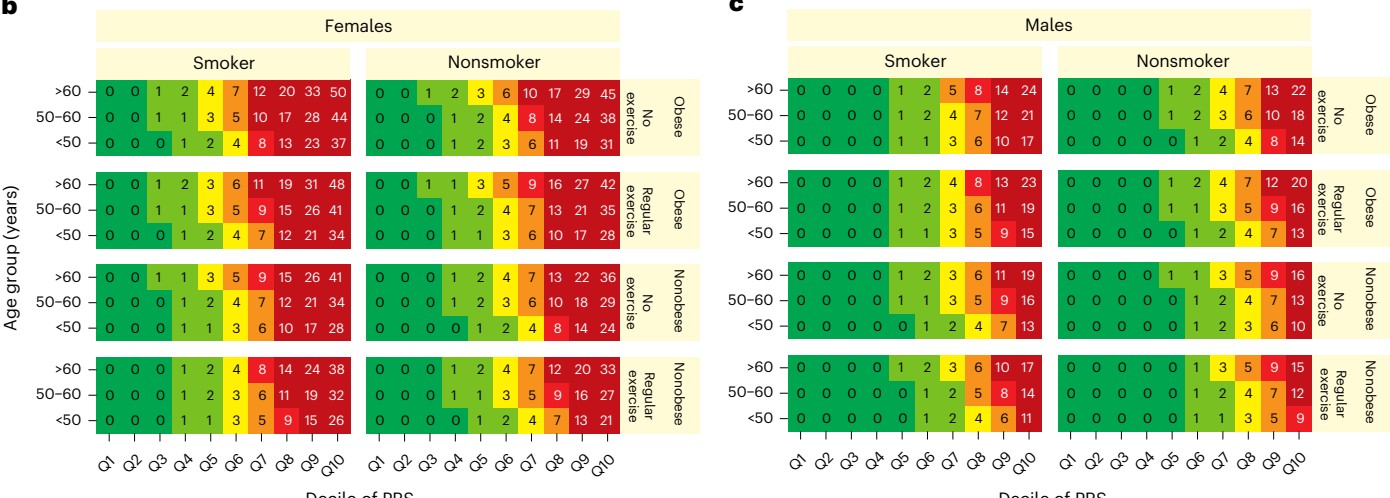

**Fig. 4 | Stratifying hypothyroidism risk using lifestyle characteristics and polygenic risk in the UKB. a,** Risk for incident hypothyroidism according to different lifestyle characteristics and categories. Data are presented as HR point estimates ±95% CIs, derived from two-sided Cox proportional hazards models, adjusted for age, sex and PCs. The center of each error bar represents the mean HR estimate. **b,c,** Ten-year risk of hypothyroidism, stratified by sex (**b**, females; **c**, males), age group, obesity status (BMI > 30), exercise regularity (yes/no), smoking status (yes/no) and divisions within the PRS across ten deciles (Q1–10).

hypothyroidism (OR = 2.01 per s.d. increase in PRS, 95% CI = 1.99–2.03, $P = 2.3 \times 10^{-2790}$). Risk increased markedly at the extremes of the PRS distribution (Fig. 2a), with ORs of 4.14 (95% CI = 3.94–4.34), 7.49 (95% CI = 6.89–8.15) and 14.10 (95% CI = 11.44–17.38) for individuals at the upper 10th, 1st and 0.1th percentiles, respectively, compared to the 5th decile. We found a similar effect estimate in the GESUS cohort (OR = 2.0 per s.d. increase in PRS 95% CI = 1.85–2.17, $P = 1.61 \times 10^{-66}$). We next evaluated the predictive ability of the PRS relative to established risk factors[6]. Relative to a model with age, sex and principal components (PCs), the PRS yielded the largest change in area under the curve (ΔAUC) of 7.2% (95% CI = 6.7–7.6), which exceeded the impact of all other risk factors (Fig. 2b). Integrating all non-genetic risk factors into a model resulted in a ΔAUC of 0.5% (95% CI = 0.4–0.7), and a model including all risk factors (including the PRS) resulted in a ΔAUC of 7.8% (95% CI = 7.3–8.2; AUC = 0.70). Anti-TPO is a strong predictor of autoimmune hypothyroidism[22]. In the GESUS cohort, we identified 5,452 individuals with TSH, fT4 and anti-TPO measurements that were free of hypothyroidism at baseline. Of these, 431 were anti-TPO positive ( > 100 U ml⁻¹). A model including age, sex and PCs yielded an AUC of 0.634 (95% CI = 0.589–0.679). A model including thyroid hormones and anti-TPO increased AUC further to 0.849 (95% CI = 0.810–0.889). By adding the PRS to the latter model, risk prediction improved significantly, increasing the AUC to 0.859 (95% CI = 0.821–0.897, P for difference = 0.03; Supplementary Table 20). For individuals who were anti-TPO negative, the PRS was able to capture residual disease risk. Anti-TPO-negative individuals in the top 10% of the PRS distribution had a nearly twofold increased risk (hazard ratio (HR) = 1.97, 95% CI = 1.06–3.68, P = 0.033) of developing hypothyroidism compared to those in the bottom 90% of the distribution.

## Disease progression in SCH

The clinical course of individuals with SCH is difficult to predict[5]. We identified 8,114 individuals from UKB primary care data with biochemically defined SCH and investigated whether the PRS could identify individuals who are more or less likely to progress to overt disease. Compared to individuals with intermediate polygenic risk (>10th to 90th percentiles), individuals with high polygenic risk (>90th percentile) had an HR of 1.43 (95% CI = 1.37–1.61) for progressing to overt hypothyroidism, while low risk individuals (>10th percentile) had an HR of 0.76 (95% CI = 0.65–0.88). On the absolute scale, this risk translated to a 10.2% higher 10-year conversion rate for high-risk individuals (39.3%, 95% CI = 35.9–42.7%) and a 6.6% lower 10-year conversion rate for low risk individuals (22.5%, 95% CI = 19.6–25.4%) compared to those in the intermediate risk group (29.1%, 95% CI = 28.0–30.3%; Fig. 3).

## Disease risk stratified by lifestyle factors and genetic risk

We investigated the relationship between hypothyroidism risk and lifestyle categories using a four-point scoring system based on body mass index (BMI), exercise, smoking and dietary habits. We found that healthy lifestyle characteristics were associated with a reduced risk of hypothyroidism. As expected, individuals without obesity had lower risk (HR = 0.71, 95% CI = 0.68–0.74) compared to obese individuals[23]. Contrary to previous findings, we found that nonsmokers had a lower risk (HR = 0.79, 95% CI = 0.75–0.84) than did current smokers[24]. Overall, adherence to a healthy lifestyle corresponded to an HR of 0.83 (95% CI = 0.79–0.87), while an unhealthy lifestyle corresponded to an HR of 1.26 (95% CI = 1.16–1.35; Fig. 4a and Supplementary Table 21). Finally, we explored the interplay between the PRS and lifestyle factors to identify individuals at extreme disease risk (Fig. 4b,c). The 10-year risk was

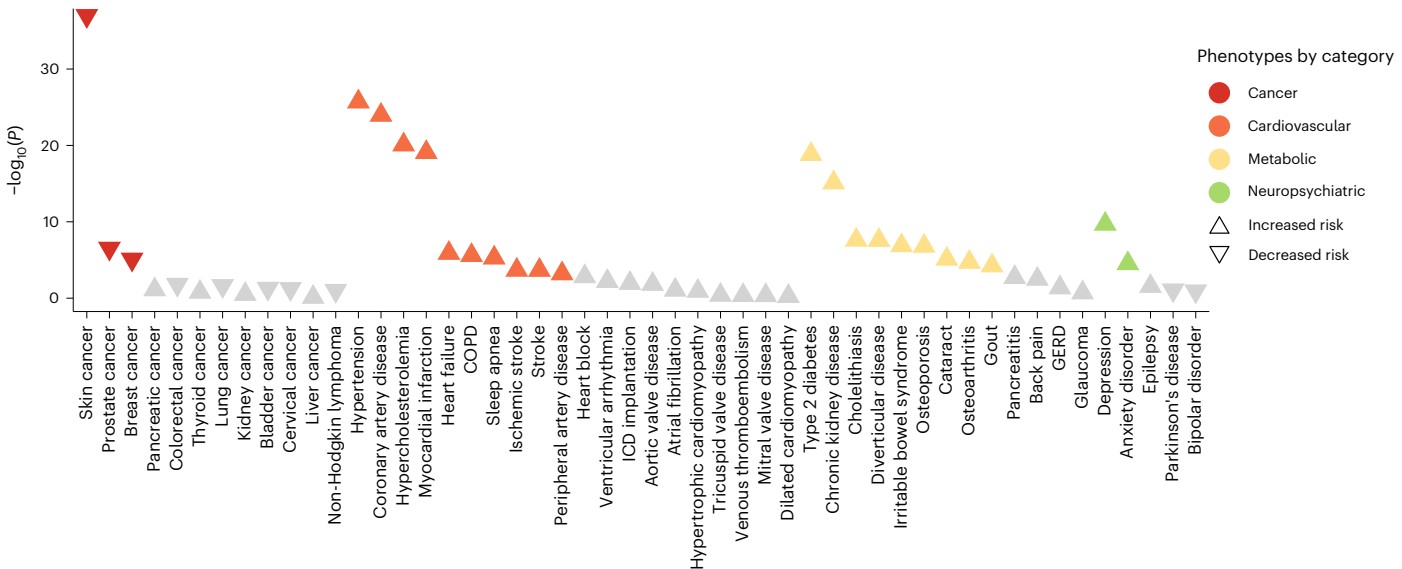

**Fig. 5 | Phenome-wide associations between the hypothyroidism PRS and cancer and cardiometabolic phenotypes in the UKB.** The figure shows associations between the hypothyroidism PRS and 50 binary disease outcomes. OR reflect the change in disease risk per 1 s.d. increase in the PRS, estimated using logistic regression models adjusted for age, sex and four PCs. *P*-values were calculated using two-sided Wald tests. Each colored triangle indicates a significant association after Bonferroni correction ($P < 0.001$, that is, 0.05/50). Upward-pointing triangles indicate increased risk and downward-pointing triangles indicate decreased risk.

greater for women, with the highest risk observed along the PRS axis. Risk increased with accumulating risk factors and higher polygenic risk, where the highest 10-year risk (50%) was observed for women above the age of 60 years, with a PRS in the >90th percentile of the distribution, who were obese, smokers and did not exercise regularly.

## Associations with cancer and cardiometabolic phenotypes

We investigated the relationship between the hypothyroidism PRS and 50 phenotypes, including common malignancies and cardiometabolic traits in UKB. We found that a higher PRS was associated with a lower risk of skin (OR = 0.92 per s.d. increase in PRS, $P = 7.7 \times 10^{-37}$), prostate (OR = 0.94 per s.d. increase in PRS, $P = 2.2 \times 10^{-6}$) and breast cancer (OR = 0.95 per s.d. increase in PRS, $P = 6.0 \times 10^{-5}$). We also found that a higher PRS was associated with an increased risk of several cardiometabolic diseases, including CAD (OR = 1.06 per s.d. increase in PRS, $P = 1.1 \times 10^{-24}$), chronic kidney disease (OR = 1.06 per s.d. increase in PRS, $P = 7.7 \times 10^{-16}$) and type 2 diabetes (OR = 1.05 per s.d. increase in PRS, $P = 1.5 \times 10^{-19}$; Fig. 5 and Supplementary Table 22).

## Discussion

In this study, we present a comprehensive genetic evaluation of thyroid hormone deficiency through GWAS meta-analyses of hypothyroidism and thyroid hormones. Our findings confirm and extend the understanding of the polygenic and complex nature of hypothyroidism, linking 350 genetic loci to this disease. By linking genetic loci to immune-related cells and circulating inflammation markers, we characterized potential mediators of disease. Using gene-prioritization methods, we identified putative genes with known roles in autoimmunity, which aligns with the main etiology in iodine-sufficient areas of the world[25]. We showed that using a hypothyroidism PRS could potentially improve the diagnostic accuracy in thyroid hormone deficiency, a condition fraught with diagnostic challenges.

We highlight associations with inflammatory markers, which may provide insight into inflammatory pathogenic mechanisms[26]. We emphasize four risk-mitigating variants (missense−rs149007883 in *NFKBIZ*, rs34536443 in *TYK2*; intronic−rs13181561 in *STING1*, rs113473633 in *NKFB1*) in genes encoding critical regulators of immune system function[27-30]. These variants were associated with lower levels of inflammatory mediators, including IFN-γ, CXCL10 and CXCL9, that make up a crucial pathway in the activation and recruitment of immune cells[18,19]. This is proposed to be a central pathogenic pathway in many autoimmune diseases, including vitiligo, psoriasis and psoriatic arthritis, which are closely linked to hypothyroidism. This aligns with the increased expression of IFN-γ in the serum and thyroid tissue of patients with autoimmune hypothyroidism, which is proposed to mediate thyroid hormone deficiency through lymphocyte infiltration and the exposure of thyrocytes to proinflammatory cytokines[4,31-33]. Further investigation into key signaling pathways, such as those highlighted, may be critical for understanding the mechanisms underlying disease pathogenesis[34].

Human genetic evidence has been acknowledged as an important predictor of success in drug development programs[34]. We observed converging evidence linking both common and rare PAVs in *TYK2* and *ZAP70* with reduced risk of hypothyroidism. Although the therapeutic potential of inhibiting TYK2 has been used in multiple autoimmune diseases, its potential in mitigating hypothyroidism risk remains largely unexplored[35,36]. This could suggest a strategic direction for drug repurposing. Similarly, inhibiting the protein product of *ZAP70*, which is also essential for T-cell signaling, has been shown to have anti-inflammatory properties *in vitro* and to be effective in treating psoriasis in mice[37]. Given its similar pathway and risk reduction profile, *ZAP70* has also emerged as a candidate for further research in the context of managing autoimmune diseases.

Due to the highly polygenic nature of hypothyroidism, we developed a PRS from more than 116,000 hypothyroidism cases to address diagnostic challenges in thyroid hormone deficiency. An estimated 0.5% of individuals with undiagnosed hypothyroidism may reflect individuals who do not seek medical attention for gradually developing nonspecific symptoms. The use of a PRS to identify individuals at greater risk could reduce the burden of undiagnosed thyroid failure. Specifically, in the top 1% and 0.1% of the PRS distribution, individuals exhibited a more than 7-fold and 14-fold risk, respectively, when compared to the middle decile. These risks are substantially greater than those observed for other complex traits[38] and for known monogenic causes of hypothyroidism. Using two different validation cohorts, we were able to show that the PRS outperformed an array of clinical

hypothyroidism risk factors but also improved risk prediction beyond that of thyroid hormones and anti-TPO.

The prevalence of thyroid hormone testing in clinical practice inevitably leads to a substantial number of patients being diagnosed and treated for SCH[5]. Early treatment is beneficial for preventing the progression to overt hypothyroidism and mitigating the risk of associated cardiovascular morbidity[39]. However, the clinical course of SCH to overt disease is unpredictable and relies on vague and nonspecific symptoms. We demonstrated that the PRS could identify individuals at high and low risk of progression from SCH to overt disease. If genotyping becomes a standard of care, PRS may guide clinicians in selecting patients who are more or less likely to progress from one disease state to another. Consequently, the clinical approach could shift to a genotype-guided biochemical assessment, rather than relying solely on nonspecific symptoms to guide testing. Also, we were able to show that by combining the PRS with easily accessible lifestyle factors, we could identify individuals with a 10-year risk of 50%. These accumulated risk factors are comparable to conventional risk factors investigated in The Wickham Study, where women with elevated TSH (>6 mU l$^{-1}$) and positive anti-TPO had an annual progression rate of 4.3%[40]. Collectively, our findings underscore the potential of using genetic risk stratification to guide personalized risk assessment and prevention strategies for hypothyroidism.

Numerous observational studies have linked hypothyroidism to increased cardiovascular morbidity[1,3,4]. Using a phenome-wide association study approach, we found that the hypothyroidism PRS associated with a range of cardiometabolic diseases, atherosclerotic disease, chronic kidney disease and type 2 diabetes. This implies the need for a more focused approach to monitoring cardiovascular risk factors and diseases in individuals with hypothyroidism. Furthermore, we found significant associations between genetically predicted higher hypothyroidism risk and lower risk of breast, prostate and skin cancer, supporting the findings reported by several observational studies[41,42]. The association between the PRS and breast cancer aligns with that of a recent GWAS of thyroid function[11]. Interestingly, we found no association between hypothyroidism risk and thyroid cancer, despite previous GWASs showing an association between higher TSH levels and lower risk of thyroid cancer[11,16]. Whether the observed associations with specific cancers reflect shared pathways, where augmented immunosurveillance leads to both disease and, conversely, mitigates the risk of specific cancers, will require additional investigation.

This study has several limitations. First, the analysis was limited to individuals of European ancestry, which restricts the generalizability of our findings to other ancestries. Second, we relied on data from cohorts, where the phenotype definition was based on self-reported diagnoses, such as those from 23andMe, or on summary statistics with predefined phenotypes, which limited our ability to further refine the phenotype definitions. This may have introduced some degree of heterogeneity.

In conclusion, we found 350 genomic risk loci for hypothyroidism, underscoring the highly polygenic nature of this disease. Leveraging this insight, we developed a PRS that could identify individuals at high risk of developing disease in the general population and also predict the clinical course of subclinical disease. Our findings represent a step forward in the genetic understanding and clinical management of hypothyroidism, broadening the perspective for use in personalized medicine.

## Online content

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

¹Department of Cardiology, Rigshospitalet, Copenhagen University Hospital, Copenhagen, Denmark. ²Cardiac Functional Genomics, Department of Biomedical Sciences, University of Copenhagen, Copenhagen, Denmark. ³Laboratory for Molecular Cardiology, Department of Cardiology, Rigshospitalet, Copenhagen University Hospital, Copenhagen, Denmark. ⁴deCODE genetics/Amgen, Inc., Reykjavik, Iceland. ⁵Department of Clinical Medicine, Faculty of Health and Medical Sciences, University of Copenhagen, Copenhagen, Denmark. ⁶Department of Nephrology and Endocrinology, Rigshospitalet, Copenhagen University Hospital, Copenhagen, Denmark. ⁷Estonian Genome Centre, Institute of Genomics, University of Tartu, Tartu, Estonia. ⁸Department of Cardiology, Copenhagen University Hospital, North Zealand, Hillerød, Denmark. ⁹Department of Endocrinology, Copenhagen University Hospital, Bispebjerg and Frederiksberg, Copenhagen, Denmark. ¹⁰Department of Laboratory Medicine, Boston Children's Hospital, Harvard Medical School, Boston, MA, USA. ¹¹Department of Clinical Biochemistry, Zealand University Hospital, Køge, Denmark. ¹²Department of Endocrinology, Herlev and Gentofte Hospital, University of Copenhagen, Herlev, Denmark. ¹³Department of Clinical Immunology, Aarhus University Hospital, Aarhus, Denmark. ¹⁴Department of Clinical Medicine, Health, Aarhus University, Aarhus, Denmark. ¹⁵Department of Clinical Immunology, Odense University Hospital, Odense, Denmark. ¹⁶Department of Clinical Immunology, Aalborg University Hospital, Aalborg, Denmark. ¹⁷Statens Serum Institut, Copenhagen, Denmark. ¹⁸Novo Nordisk Foundation Center for Protein Research, Faculty of Health and Medical Sciences, University of Copenhagen, Copenhagen, Denmark. ¹⁹Department of Public Health, University of Copenhagen, Copenhagen, Denmark. ²⁰Department of Clinical Immunology, Rigshospitalet, Copenhagen University Hospital, Copenhagen, Denmark. ²¹Department of Clinical Immunology, Zealand University Hospital, Køge, Denmark. ²²Faculty of Medicine, School of Health Sciences, University of Iceland, Reykjavik, Iceland. ²³Department of Molecular Medicine, Aarhus University Hospital, Aarhus University, Aarhus, Denmark. ²⁴Department of Clinical Biochemistry, Aarhus University Hospital, Aarhus, Denmark. ³¹These authors contributed equally: Henning Bundgaard, Jonas Ghouse. *Lists of authors and their affiliations appear at the end of the paper. ✉e-mail: soeren.albertsen.rand.02@regionh.dk; jonasghouse@clin.au.dk

## DBDS Genomic Consortium

**Karina Banasik²⁵, Jakob Bay²¹, Andrea Barghetti²⁰, Mette Skou Bendtsen²⁰, Jens Kjærgaard Boldsen¹³, Søren Brunak¹⁸,¹⁹, Nanna Brøns²⁰, Alfonso Buil Demur²⁶, Johan Skov Bundgaard²⁰, Lea Arregui Nordahl Christoffersen²¹, Maria Didriksen²⁰, Khoa Manh Dinh¹³, Joseph Dowsett²⁰, Christian Erikstrup¹³,¹⁴, Josephine Gladov¹³,**

**Daniel F. Gudbjartsson**[4], **Thomas Folkmann Hansen**[27], **Dorte Helenius Mikkelsen**[26], **Lotte Lindhede**[13], **Henrik Hjalgrim**[28], **Jakob Hjorth von Stemann**[20], **Bitten Aagaard Jensen**[16], **Kathrine Kaspersen**[13], **Bertram Dalskov Kjerulff**[13], **Lisette Kogelman**[27], **Mette Kongstad**[21], **Susan Mikkelsen**[13], **Christina Mikkelsen**[20], **Line Hjorth Sjernholm Nielsen**[13], **Janna Nissen**[20], **Mette Nyegaard**[29], **Sisse R. Ostrowski**[5,20], **Frederikke Byron Pedersen**[20], **Ole B. Pedersen**[5,21], **Liam James Elgaard Quinn**[21], **Thorunn Rafnar**[4], **Palle Duun Rohde**[29], **Klaus Rostgaard**[28], **Andrew Joseph Schork**[26], **Michael Schwinn**[20], **Erik Sørensen**[20], **Kari Steffanson**[4,22], **Hreinn Stefansson**[4], **Jacob Træholt**[20], **Unnur Thorsteinsdottir**[4], **Mie T. Bruun**[15], **Henrik Ullum**[17], **Thomas Werge**[26] & **David Westergaard**[25]

[25]Department of Obstetrics and Gynaecology, Copenhagen University Hospital, Hvidovre Hospital, Copenhagen, Denmark. [26]Institute of Biological Psychiatry, Mental Health Centre, Sct. Hans, Copenhagen University Hospital, Roskilde, Denmark. [27]Danish Headache Center, Department of Neurology, Copenhagen University Hospital, Rigshospitalet-Glostrup, Copenhagen, Denmark. [28]Danish Cancer Society Research Center, Copenhagen, Denmark. [29]Department of Health Science and Technology, Faculty of Medicine, Aalborg University, Aalborg, Denmark.

## Estonian Biobank Research Team

**Andres Metspalu**[7], **Lili Milani**[7], **Tony Esko**[7], **Reedik Magi**[7], **Mait Metspalu**[7], **Mari Nelis**[7] & **Georgi Hudjashov**[7]

## 23andMe Research Team

**Stella Aslibekyan**[30], **Adam Auton**[30], **Robert K. Bell**[30], **Katelyn Kukar Bond**[30], **Zayn Cochinwala**[30], **Sayantan Das**[30], **Kahsaia de Brito**[30], **Emily DelloRusso**[30], **Chris Eijsbouts**[30], **Sarah L. Elson**[30], **Chris German**[30], **Julie M. Granka**[30], **Alan Kwong**[30], **Yanyu Liang**[30], **Keng-Han Lin**[30], **Matthew H. McIntyre**[30], **Shubham Saini**[30], **Anjali J. Shastri**[30], **Jingchunzi Shi**[30], **Suyash Shingrapure**[30], **Qiaojuan Jane Su**[30], **Vinh Tran**[30], **Joyce Y. Tung**[30], **Catherine H. Weldon**[30] & **Wanwan Xu**[30]

[30]23andMe, Inc., Sunnyvale, CA, USA.

# Methods

## Ethics statement

This research complied with all ethical regulations and was conducted in accordance with the principles of the Declaration of Helsinki. All contributing studies received approval from the appropriate regional or institutional research ethics committees. For most cohorts, participants provided written informed consent before inclusion. For CHB participants, written informed consent was not obtained, but in accordance with Danish legislation, participants were informed about the use of residual biological material for research purposes and provided with the option to opt out. Details of ethics approvals and consent procedures for individual cohorts are provided in the Supplementary Note.

## Cohorts, association testing and meta-analysis

We used four cohorts for discovery (CHB-CID/DBDS, UKB, FinnGen Freeze 10 and 23andMe). EstBB and deCODE genetics were used for validation. Cases were defined using International Classification of Diseases (ICD)-10 codes E03.8/E03.9/E06.3, ICD-9 codes 244.8/244.9 or claimed prescription of thyroid hormone substitution therapy using Anatomical Therapeutic Chemical Classification code H03A. In UKB, in addition to electronic health registries, we included individuals self-reporting hypothyroidism or use of thyroid hormone substitution as cases. Individuals with hyperthyroidism (E05(0-9)) were excluded and we otherwise used thyroid disease-free controls, excluding individuals with the following ICD-10 codes: E0(1-2), E03(0-5) and E0(4-7). In 23andMe, cases were defined based on self-reported diagnoses of hypothyroidism, elevated thyroid-stimulating hormone or taking levothyroxine. Controls were individuals who reported no other thyroid-related disorders[43]. Details on genotyping, imputation and quality control are provided in the Supplementary Note and Supplementary Table 23. Using CHB-CID/DBDS, UKB primary care data (the subset allowed for non-COVID research) and previously published data, we meta-analyzed GWASs for TSH and fT4. We used the first non-missing sample value that was within the reference range. The results of individual thyroid function tests were inverse normalized. Individuals who were either on thyroid drugs or had undergone thyroid surgery before the thyroid function tests were excluded. In the UKB primary care data, thyroid hormone measurements were captured using Read2 and Read3 codes, while drug and operation codes were recorded using dm+d and OPCS-4 codes, respectively. In the CHB-DBDS, thyroid hormones were captured using NPU codes, drugs using ATC codes and surgical procedures using procedure codes (Supplementary Table 24). Each dataset underwent initial quality control (QC), imputation, post-imputation QC and logistic regression models were used for the hypothyroidism GWAS and linear regression for the thyroid hormones. All models were adjusted for age, sex and PCs. In postregression QC, we removed variants with an imputation quality score <0.6, minor allele count (MAC) < 6 or absolute $\beta$ or s.e. >10. We meta-analyzed datasets using METAL[44], using the fixed-effect inverse variance weighted method. To evaluate genomic inflation, we calculated the genomic inflation factor ($\lambda_{GC}$) and the LDSC-intercept using LD scores calculated in the HapMap3 CEU population (Supplementary Table 1). We observed signs of inflation in FinnGen$_{Hypo-GWAS}$ ($\lambda_{GC} = 1.40$, LDSC-intercept = 1.21), CHB-CID/DBDS$_{TSH-GWAS}$ ($\lambda_{GC} = 1.35$, LDSC-intercept = 1.19) and UKB$_{TSH-GWAS}$ ($\lambda_{GC} = 1.24$, LDSC-intercept = 1.1) and accounted for potential bias by correcting the GWAS s.e. by the square root of the LDSC-intercept[45]. No additional genomic control was applied. Liftover between genetic builds was conducted using the R package MungeSumstats[46]

## Risk locus definition

To identify independent SNPs within each risk locus, we used LD clumping from PLINK (v1.9)[47]. We applied a 1-Mb window (--clump kb 1000) and low LD threshold (--r² 0.1) to identify independently significant SNPs. Lead SNPs were independent SNPs with the lowest $P$ value, and a locus was defined as a ±1-Mb region around each lead SNP. We queried the GWAS-catalog[48] (on 19 April 2024) for known phenotypic associations with either the lead SNP or variants located ±1 Mb of the lead SNP. We considered a risk locus new if no genome-wide significant association ($P < 5 \times 10^{-8}$) with hypothyroidism or the use of thyroid hormone replacement therapy had been reported previously.

## Heritability

Variance in hypothyroidism risk and levels of fT4 and TSH explained by common SNPs were estimated using LD-adjusted kinships (LDAK) SumHer BLD-LDAK model[49]. We used the precomputed tagging files internal to SumHer, and for hypothyroidism, assessed the heritability on a liability scale (correcting for sample and population prevalence).

## Association with blood cell traits and inflammatory proteins in the UKB

Investigating the genetic imprint on the immunophenotype of hypothyroidism may help identify key functions and interactions involved in hypothyroidism risk. First, we assessed 10 blood cell counts, including basophil, eosinophil, neutrophil, monocyte, platelet, reticulocyte, high light-scattered reticulocyte, lymphocyte, red and white blood cell counts, along with C-reactive protein levels. We obtained these measurements from Europeans in the UKB and subsequently rank-inversely normalized each trait. Next, we tested the association between lead variants and blood cell traits using linear regression adjusted for age, sex and four PCs. Additionally, we used proteomics data from the UKB Pharma Proteomics Project[50]. We assessed the effect of lead variants on 90 inflammatory proteins, which were identified as immune-mediated drivers[17]. We set the threshold for multiple testing at $P = 1.41 \times 10^{-6}$ (0.05/101 traits x 350 variants).

## Gene mapping

To identify and prioritize candidate genes at each locus (±1 Mb of the lead variant), we used the following five methods:

1. Coding variants—we investigated whether lead variants or proxy variants ($r^2 > 0.8$) were annotated as coding variants using the variant effect predictor[51].
2. V2G—we used the V2G algorithm provided by Open Targets Genetics (https://genetics.opentargets.org/), which scores and assigns each variant to a gene based on aggregated evidence from splice, expression, and protein quantitative trait loci (sQTL, eQTL and pQTL, respectively), chromatin interactions, in silico prediction and distance to transcript sites.
3. PoPS—we used a similarity-based gene-prioritization tool integrating GWAS summary data, gene expression data, biological pathways data and protein-protein interaction data from over 50,000 features[52]. The analysis involved the following three steps: (1) computing gene-level association data and gene-gene correlations using MAGMA[53] with LD estimated from 1000 Genomes European data, (2) running enrichment analysis for gene features listed at https://github.com/FinucaneLab/gene_features using MAGMA and (3) calculating PoPS score for each gene by fitting a joint model for enrichment of all resulting features. Genes with a PoPS score in the top 10% of the distribution were prioritized as putative causal genes.
4. Mendelian disease enrichment—we used MendelVar (https://mendelvar.mrcieu.ac.uk/) to detect intersections between hypothyroidism loci and Mendelian disease genes, providing valuable clues for gene prioritization. We selected variants located within ±1 Mb of each lead variant and used the 1000 Genomes Europeans as a reference panel. Genes were annotated if they were identified with the following keywords: 'thyroid', 'immune' or 'immuno' in OMIM disease descriptions, or Human Phenotype Ontology/Disease Ontology descriptions.

5.  TWAS with colocalization—using FUSION[54], we performed TWASs using hypothyroidism summary data to investigate the relationship between the risk loci and effects on gene expression in GTEx v8 datasets on hypothalamus, pituitary gland, thyroid, spleen, whole blood and pancreas. We used the internal colocalization function, which uses COLOC, to detect shared causal variants between hypothyroidism risk and gene expression[55]. We only considered eQTLs associated with hypothyroidism at $P < 2.97 \times 10^{-6}$ (0.05/16,841 genes tested). Finally, we report the posterior probability of colocalized associations (PP4), which show evidence of a shared causal variant found in both GWAS and functional associations. If a hypothyroidism risk locus (sentinel variant ±1 Mb) harbored a gene with PP4 > 0.75, we considered this a mediator of hypothyroidism and evidence of gene mapping.

Genes with two or more lines of evidence for gene mapping were investigated in a Gene Ontology enrichment analysis using the R package clusterProfiler[56].

### Rare protein-truncating variants in prioritized hypothyroidism genes
The convergence in disease risk between common and rare truncating variants can pinpoint causal genes. This can offer insight into disease pathophysiology and potentially guide drug discovery or repurposing. Using published summary statistics from a whole-exome sequencing burden analysis of hypothyroidism[20], we compared the convergence between common and rare variant effects in genes that had at least two lines of mapping evidence at FDR < 0.05.

### PRS derivation and validation
We generated the hypothyroidism PRS from a meta-analysis of CHB-CID/DBDS, FinnGen, EstBB, deCODE genetics and 23andMe, comprising 116,470 cases and 1,164,733 controls. PRS weights were calculated using PRS-CS[57] with an LD reference panel from the UKB. We validated the PRS in UKB, where individuals with ICD-10 E05(0-9) were removed to mitigate enrichment for participants with hyperthyroidism amongst hypothyroidism cases. In UKB, associations with hypothyroidism were first reported on a continuous scale. Next, we tested the association by deciles, and at the extremes of the PRS (99th and 99.9th percentiles) using logistic regression models adjusted for age, sex and four PCs. We assessed the predictive performance of the PRS relative to known clinical risk factors (for example, BMI and selected autoimmune diseases; Supplementary Table 25) by calculating the AUC using the R package pROC[58]. For each risk factor, the change in the AUC (ΔAUC) was compared to that of a model consisting of age, sex and four PCs. We evaluated prevalent risk factors (that is, events before the baseline date) and tested the predictive performance for incident hypothyroidism cases (events after the baseline date). Significant differences between the prediction models were tested using DeLong's test for correlated ROC curves. The PRS was also evaluated in the Danish General Suburban Population Study (GESUS). This was a population-based cohort study in which 21,205 adults were recruited between 2010 and 2013. At baseline, participants underwent physical examination, completed a questionnaire and had blood samples drawn. Individuals with ICD-10 E05(0-9) were excluded from the analysis. First, we evaluated the PRS association with hypothyroidism on a linear basis and then evaluated the predictive performance of the PRS relative to thyroid hormones and anti-TPO positivity. Anti-TPO was measured on Kryptor assays. Values greater than 60 U ml⁻¹ are considered positive in Denmark[59], and we chose a more conservative cutoff of 100 U ml⁻¹ for anti-TPO positivity to avoid misclassification of individuals with autoimmunity[60]. We compared the AUCs of models including sex, age and four PCs with stepwise addition of thyroid hormones, anti-TPO positivity and the PRS for hypothyroidism.

### Disease progression in SCH patients
In UKB primary care data ($n \approx 245{,}000$), we defined individuals with SCH as having TSH levels greater than 4 mU l⁻¹ and fT4 levels between 8 and 14.5 pmol l⁻¹ using Read2 and Read3 codes. We only considered biochemical measurements available after the date of enrollment in the UKB to avoid immortal time bias. Before the date for SCH, we excluded individuals with a history of thyroid cancer, hyperthyroidism and hypothyroidism using ICD-10, Read2 and Read3 codes and individuals taking thyroid hormone substitution as indicated by dm+d codes (Supplementary Table 26). The PRS was categorized into the following three groups: (1) ≤10th percentile, representing low-risk individuals; (2) >10th and <90th percentiles, representing the general population and (3) ≥90th percentile, representing high-risk individuals. We used Cox regression models to compute HRs for risk of progression to overt hypothyroidism. Individual follow-up ended in the case of an event sampled following the date of SCH from electronic health records (defined by ICD-10 E03.8/E03.9/E06.3 or Read2/Read3 codes indicative of autoimmune myxedema/Hashimoto's thyroiditis), death or end of follow-up, whichever occurred first. Models were adjusted for age, sex and four PCs. Absolute risks were calculated using the Aalen-Johansen estimator, which takes the competing risk of death into account.

### Relationship between polygenic risk and lifestyle
Lifestyle factors such as obesity are part of the phenotypic spectrum of hypothyroidism but can also increase the risk of hypothyroidism[23,24]. Since inherited risk can be perceived as deterministic, we investigated whether adherence to a healthy lifestyle could offset genetic risk. Using UKB questionnaire data, we created a lifestyle scoring system with points awarded for healthy characteristics[61]:

1.  No obesity (BMI < 30 kg m⁻²)
2.  Regular exercise (≥15 metabolic equivalent task hours per week)
3.  Nonsmokers
4.  Healthy diet, meeting at least three criteria:

    a.  ≥3 fruit servings per day.
    b.  ≥12 teaspoons of vegetables per day.
    c.  ≥2 weekly servings of oily fish.
    d.  ≤1 weekly serving of processed meat.
    e.  ≤2 weekly servings of red meat.

A healthy lifestyle ranged from 3–4 points, intermediate 2 points and unhealthy 0–1 points. We analyzed the associations between lifestyle factors and hypothyroidism using Cox regression, adjusting for age at inclusion, sex and four PCs using the R package survival[62]. We also constructed a risk chart, displaying 10-year absolute risk of hypothyroidism for different combinations of age, sex, lifestyle characteristics and PRS deciles.

### PRS correlation with selected malignancies, cardiometabolic and neuropsychiatric traits
We conducted a phenome-wide association study between the hypothyroidism PRS and 50 common diseases (including 12 malignant, 20 cardiovascular, 13 metabolic and 4 neuropsychiatric traits). We defined phenotypes in the UKB using a combination of ICD-9 and ICD-10 codes, cause of death registry and Office of Population Censuses and Surveys (OPCS-4; Supplementary Table 27). We tested the association between the PRS and individual phenotypes using logistic regression adjusted for age, sex and four PCs. We restricted analyses of cervical and breast cancer to females and prostate cancer to males. We set the threshold for multiple testing at $P < 0.001$ (0.05/50 traits).

### Reporting summary
Further information on research design is available in the Nature Portfolio Reporting Summary linked to this article.

## Data availability

GWAS summary statistics from the meta-analysis of hypothyroidism (excluding 23andMe), TSH and T4 are publicly available at the GWAS Catalog under accession IDs GCST90572791 (https://ftp.ebi.ac.uk/pub/databases/gwas/summary_statistics/GCST90572001-GCST90573000/GCST90572791/), GCST90572789 (https://ftp.ebi.ac.uk/pub/databases/gwas/summary_statistics/GCST90572001-GCST90573000/GCST90572789/) and GCST90572790 (https://ftp.ebi.ac.uk/pub/databases/gwas/summary_statistics/GCST90572001-GCST90573000/GCST90572790/). The corresponding hypothyroidism PRS (excluding 23andMe) is available at the PGS Catalog under accession ID PGS005218 (https://www.pgscatalog.org/publication/PGP000733/). The full GWAS summary statistics for the 23andMe discovery dataset will be made available through 23andMe to qualified researchers under an agreement with 23andMe that protects the privacy of the 23andMe participants. Please visit https://research.23andme.com/collaborate/#dataset-access/. UKB individual-level data are accessible upon application via the UKB (https://www.ukbiobank.ac.uk/). FinnGen summary statistics are publicly available following registration at: https://www.finngen.fi/en/access_results. Data from the UKB Pharma Proteomics Project (UKB-PPP) are available through Synapse (https://www.synapse.org/#!Synapse:syn51365301). GTEx v8 eQTL data can be accessed at: https://gtexportal.org/home/datasets. Individual-level data are not publicly available due to restrictions imposed by participant consent and local ethics review boards.

## Code availability

The following software and packages were used for data analyses:

PLINK 1.9 (https://www.cog-genomics.org/plink/1.9/)

PLINK 2.0 (https://www.cog-genomics.org/plink/2.0/)

METAL v.2011-03-25 (https://genome.sph.umich.edu/wiki/METAL_Documentation)

LDAK v.2023-07-01 (https://dougspeed.com/)

PoPS v.0.2 (https://github.com/Finucanelab/pops)

FUSION v.2022/02/01 (http://gusevlab.org/projects/fusion/)

COLOC v.5.2.3 (https://cran.r-project.org/web/packages/coloc/index.html)

LD Score Regression v.l .0.1 (https://github.com/bulik/ldsc)

PRS-CS v.2021-06-04 (https://github.com/getian107/PRScs)

REGENIE v.2.0.l (https://rgcgithub.github.io/regenie/)

R v.4.2.2 (https://www.r-project.org/)

MungeSumstats v.1.8.0 (https://www.bioconductor.org/packages/release/bioc/html/MungeSumstats.html)

pROC v.1.18.5 (https://www.rdocumentation.org/packages/pROC/versions/1.18.5)

clusterProfiler v.4.8.2 (https://bioconductor.org/packages/release/bioc/html/clusterProfiler.html)

survival v.3.6.4 (https://cran.r-project.org/web/packages/survival/index.html)

ggplot2 v.3.5.2 (https://cran.r-project.org/web/packages/ggplot2/index.html)

OpenTargets Variant2Gene v.1.1 (https://genetics-docs.opentargets.org/our-approach/data-pipeline)

Variant Effect Predictor v.111 (https://useast.ensembl.org/Tools/VEP).

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

## Acknowledgements

We acknowledge the participants and investigators of the FinnGen study. In addition, we thank the research participants and employees of 23andMe for making this work possible. This research was conducted using the UKB resource under application 43247. U.F.-R. research salary was sponsored by a grant from the Kirsten and Freddy Johansen's Fund. C. Ellervik was partly funded by the Laboratory Medicine Endowment Fund of Boston Children's Hospital, USA. This work was supported by BRIDGE—Translational Excellence Programme (NNF18SA0034956 and NNF20SA0064340 to J.G.), The John and Birthe Meyer Foundation (to M.S.O.), The Beckett Foundation (23-2-11066 to S.A.R. and 23-2-10636 to J.G.), Agnes and Knut Mørks Foundation (to U.F.-R.), Danish Health and Medicines Authority (to H.U.), the patient society: 'Stofskifteforeningen' (to M.C.K. and U.F.-R.), the

Innovation Fund Denmark (PM Heart to H.B.), NordForsk (to H.B.), Villadsen Family Foundation (to H.B.), the Arvid Nilsson Foundation (to H.B.), Sygeforsikringen 'danmark' (to M.S.O.), Estonian Research Council grants PRG1911 (to A.M.) and TK214 (to M.T.-L. and A.M.), The Hallas-Møller Emerging Investigator Novo Nordisk (NNF17OC0031204 to M.S.O.), AUFF Recruitment grant (AUFF-E-2024-7-10 to J.G.) and Novo Nordisk Foundation (NNF17OC0027594 and NNF14CC0001 to S.B. and NNF22OC0079592 and NNF17OC0031204 to M.S.O.). All human research was approved within each contributing study by the relevant institutional review board (CHB-CID/DBDS: National Committee on Health Research Ethics; deCODE: National Bioethics Committee; UKB: Northwest Multicenter Research Ethics Committee; FinnGen: The Coordinating Ethics Committee of the Hospital District of Helsinki and Uusimaa; Estonian Biobank: Estonian Committee on Bioethics and Human Research; 23andMe Salus Institutional Review Board (formerly Ethical and Independent Review Services) and the independent and external AAHRPP-accredited institutional review board; GESUS: Ethics Committee for Health Research for Region Zealand) and conducted according to the Declaration of Helsinki. All participants provided written informed consent, except for CHB-CID, where patients were informed about the opt-out possibility of having their biological specimens excluded from use in research. Since 2004, a national Register on Tissue Application (Vævsanvendelsesregistret) lists all individuals who have chosen to opt out and whose samples cannot be used for research purposes. Before initiating this study, individuals listed in the Register on Tissue Application were excluded.

## Author contributions

S.A.R., G.A., M.S.O., H.B. and J.G. conceived the study. S.A.R., V.T., M.T.-L., H.E.P., B.N., S.S. and J.G. performed analyses in the respective cohorts. S.S., I.J., H.H., K.S., D.F.G., A.M., M.T.-L., H.E.P., S.R.O., O.B.P., M.S.O., H.B. and J.G. supervised analyses in their respective cohorts. S.A.R., G.A., S.S., M.S.O., H.B. and J.G. wrote the paper. S.A.R., G.A. and J.G. performed meta-analysis and created figures and tables. S.A.R., G.A., L.M.M. and J.G. performed downstream analysis and drafted the paper. L.M.M., C.Z., U.F.-R., M.C.K., H.E.P., C. Ellervik, B.N., C. Erikstrup, M.T.B., B.A.J., H.U., S.B., M.S., S.R.O., O.B.P., E.S., I.J., D.F.G., G.T., H.H., S.S. and K.S. interpreted the results and reviewed and commented on the paper.

## Competing interests

V.T., I.J., D.F.G., G.T., H.H., S.S. and K.S. are employees of deCODE genetics/Amgen. H.B. receives lecture fees from Bristol-Myers Squibb, General Electrics, Amgen, Sanfoi, Merck Sharp and Dohme. S.B. is a board member for Proscion A/S and Intomics A/S. J.G. has received lecture fee from Illumina and is a former employee of Novo Nordisk A/S. The other authors declare no competing interests.

## Additional information

**Correspondence and requests for materials** should be addressed to Søren A. Rand or Jonas Ghouse.

Dr. Jonas Ghouse

# Reporting Summary

## Statistics

For all statistical analyses, confirm that the following items are present in the figure legend, table legend, main text, or Methods section.

| n/a | Confirmed | |
|---|---|---|
| ☐ | ☒ | The exact sample size (*n*) for each experimental group/condition, given as a discrete number and unit of measurement |
| ☒ | ☐ | A statement on whether measurements were taken from distinct samples or whether the same sample was measured repeatedly |
| ☐ | ☒ | The statistical test(s) used AND whether they are one- or two-sided<br>*Only common tests should be described solely by name; describe more complex techniques in the Methods section.* |
| ☐ | ☒ | A description of all covariates tested |
| ☐ | ☒ | A description of any assumptions or corrections, such as tests of normality and adjustment for multiple comparisons |
| ☐ | ☒ | A full description of the statistical parameters including central tendency (e.g. means) or other basic estimates (e.g. regression coefficient) AND variation (e.g. standard deviation) or associated estimates of uncertainty (e.g. confidence intervals) |
| ☐ | ☒ | For null hypothesis testing, the test statistic (e.g. *F*, *t*, *r*) with confidence intervals, effect sizes, degrees of freedom and *P* value noted<br>*Give P values as exact values whenever suitable.* |
| ☒ | ☐ | For Bayesian analysis, information on the choice of priors and Markov chain Monte Carlo settings |
| ☒ | ☐ | For hierarchical and complex designs, identification of the appropriate level for tests and full reporting of outcomes |
| ☐ | ☒ | Estimates of effect sizes (e.g. Cohen's *d*, Pearson's *r*), indicating how they were calculated |

*Our web collection on statistics for biologists contains articles on many of the points above.*

## Software and code

Policy information about availability of computer code

| Data collection | No software was used to collect this data. |
|---|---|
| Data analysis | The following softwares and packages were used for data analyses:<br>PLINK 1.9 (https://www.cog-genomics.org/plink/1.9/)<br>PLINK 2.0 (https://www.cog-genomics.org/plink/2.0/)<br>METAL v2011-03-25 (https://genome.sph.umich.edu/wiki/METAL_Documentation)<br>LDAK v2023-07-01 (https://dougspeed.com/)<br>PoPS v0.2 (https://github.com/Finucanelab/pops)<br>FUSION v2022/02/01 (http://gusevlab.org/projects/fusion/)<br>COLOC v5.2.3 (https://cran.r-project.org/web/packages/coloc/index.html)<br>LD Score Regression vl .0.1 (https://github.com/bulik/ldsc)<br>PRS-CS v2021-06-04 (https://github.com/getian107 /PRScs)<br>REGENIE v2.0.l (https://rgcgithub.github.io/regenie/)<br>R v4.2.2 (https://www.r-project.org/)<br>MungeSumstats v1.8.0 (https://www.bioconductor.org/packages/release/bioc/html/MungeSumstats.html)<br>pROC (https://www.rdocumentation.org/packages/pROC/versions/1.18.5)<br>clusterProfiler v3.21 (https://bioconductor.org/packages/release/bioc/html/clusterProfiler.html)<br>survival v3.6.4 (https://cran.r-project.org/web/packages/survival/index.html)<br>ggplot2 (https://cran.r-project.org/web/packages/ggplot2/index.html)<br>OpenTargets Variant2Gene v1.1 (https://genetics-docs.opentargets.org/our-approach/data-pipeline)<br>MendelVar v05/Dec/2023 (https://mendelvar.mrcieu.ac.uk/) |

Variant Effect Predictor v111 (https://useast.ensembl.org/Tools/VEP)

For manuscripts utilizing custom algorithms or software that are central to the research but not yet described in published literature, software must be made available to editors and reviewers. We strongly encourage code deposition in a community repository (e.g. GitHub). See the Nature Portfolio guidelines for submitting code & software for further information.

# Data

Policy information about availability of data

All manuscripts must include a data availability statement. This statement should provide the following information, where applicable:
- Accession codes, unique identifiers, or web links for publicly available datasets
- A description of any restrictions on data availability
- For clinical datasets or third party data, please ensure that the statement adheres to our policy

GWAS summary statistics from the meta-analysis of hypothyroidism (excluding 23andMe), TSH and T4 are publicly available at the GWAS Catalog under accession IDs: GCST90572791, GCST90572789 and GCST90572790 (https://www.ebi.ac.uk/gwas/). The corresponding hypothyroidism polygenic risk score (excluding 23andMe) is available at the PGS Catalog under accession ID: PGS005218 (https://www.pgscatalog.org/).The full GWAS summary statistics for the 23andMe discovery data set will be made available through 23andMe to qualified researchers under an agreement with 23andMe that protects the privacy of the 23andMe participants. Please visit https://research.23andme.com/collaborate/#dataset-access/. UK Biobank individual-level data are accessible upon application via the UK Biobank (https://www.ukbiobank.ac.uk/). FinnGen summary statistics are publicly available following registration at: https://www.finngen.fi/en/access_results. Data from the UK Biobank Pharma Proteomics Project (UKB-PPP) are available through Synapse (https://www.synapse.org/#!Synapse:syn51365301). GTEx v8 eQTL data can be accessed at: https://gtexportal.org/home/datasets. Individual-level data are not publicly available due to restrictions imposed by participant consent and local ethics review boards.

# Research involving human participants, their data, or biological material

Policy information about studies with human participants or human data. See also policy information about sex, gender (identity/presentation), and sexual orientation and race, ethnicity and racism.

| | |
|---|---|
| Reporting on sex and gender | The manuscript uses the term sex when referring to biological attribute, and was determined using genetic sex where available. Sex was included as a covariate in all multivariate analyses. |
| Reporting on race, ethnicity, or other socially relevant groupings | The included studies were exclusively individuals of European ancestry. By reducing genetic variability and confounding factors that could arise from population stratification, we aimed to enhance our ability to detect true genetic associations with the phenotypes of interest. Our study's focus on Europeans is not meant to diminish the genetic diversity and significance of other populations but was a methodological decision based on the specific aims and context of our research. |
| Population characteristics | Population characteristics include age, sex, ancestry, and genetic principal components for all individuals. Details on population characteristics are provided in Supplementary Tables 1 and 23, in the Online Methods section, and in the Supplementary Note. |
| Recruitment | Recruitment information is provided in Supplementary Note. |
| Ethics oversight | All human research was approved within each contributing study by the relevant institutional review board (CHB-CID/DBDS: National Committee on Health Research Ethics; deCODE: National Bioethics Committee; UKB: Northwest Multicenter Research Ethics Committee; FinnGen: The Coordinating Ethics Committee of the Hospital District of Helsinki and Uusimaa; Estonian Biobank: Estonian Committee on Bioethics and Human Research; 23andMe, Inc: Salus IRB (formerly Ethical and Independent Review Services) and the independent and external AAHRPP-accredited Institutional Review Board (IRB); GESUS: Ethics Committee for Health Research for Region Zealand) and conducted according to the Declaration of Helsinki. All participants provided written informed consent, except for CHB-CID, where patients were informed about the opt-out possibility of having their biological specimens excluded from use in research. Since 2004, a national Register on Tissue Application (Vævsanvendelsesregistret) lists all individuals who have chosen to opt out and whose samples cannot be used for research purposes. Before initiating this study, individuals listed in the Register on Tissue Application were excluded. Additional information is provided in Supplementary Note. |

Note that full information on the approval of the study protocol must also be provided in the manuscript.

# Field-specific reporting

Please select the one below that is the best fit for your research. If you are not sure, read the appropriate sections before making your selection.

☒ Life sciences        ☐ Behavioural & social sciences        ☐ Ecological, evolutionary & environmental sciences

For a reference copy of the document with all sections, see nature.com/documents/nr-reporting-summary-flat.pdf

# Life sciences study design

All studies must disclose on these points even when the disclosure is negative.

| | |
|---|---|
| Sample size | No formal statistical sample size calculations were performed prior to the study. Instead, we included all available individuals from each |

| Sample size | contributing cohort who met the inclusion criteria and passed quality control procedures. This approach maximizes statistical power for genetic discovery. The final sample sizes are comparable to or larger than those used in previous well-powered genome-wide association studies (GWAS) for related traits. Detailed information on sample inclusion and exclusion is provided in the Supplementary Note. |
|---|---|
| Data exclusions | Within each study, samples were excluded on the basis of sample level quality control and variant level quality control. These procedures ensure the removal of poor quality genotypes, SNPs and samples. The quality filtering steps are provided in Supplementary Table 23 and in Supplementary Note. |
| Replication | All GWAS analyses were conducted independently within each cohort and subsequently meta-analyzed. The primary findings were replicated in an independent meta-analysis including non-overlapping cohorts. Prior to replication, we performed power calculations to determine our ability to detect novel hypothyroidism-associated variants at various minor allele frequencies (MAFs) and odds ratios (ORs), using a Bonferroni-corrected significance threshold of $\alpha = 0.00028$ (0.05/179 replication attempts). As shown in Supplementary Fig. 1, we had >80% power to detect variants with OR ≥1.08 and MAF >0.02. Details of replication are documented in the manuscript and Supplementary Fig. 1 and Supplementary Table 8, 9, and 10. |
| Randomization | Randomization was not applicable to this study because it is an observational genetic association study based on pre-existing biobank data. Participants were not assigned to groups or interventions; instead, genetic and phenotypic data were analyzed as collected. The goal was to identify naturally occurring genetic variants associated with hypothyroidism and related traits, which does not require or permit random allocation. |
| Blinding | Blinding was not relevant to this study because it was an observational analysis of existing genotype and phenotype data from biobanks. No interventions were administered, and no subjective assessments were performed by investigators. Genotyping, quality control, and association analyses were conducted using automated pipelines without investigator influence on group assignment or outcome assessment. Therefore, the risk of bias typically mitigated by blinding was not present in this study design. |

# Reporting for specific materials, systems and methods

We require information from authors about some types of materials, experimental systems and methods used in many studies. Here, indicate whether each material, system or method listed is relevant to your study. If you are not sure if a list item applies to your research, read the appropriate section before selecting a response.

## Materials & experimental systems

| n/a | Involved in the study |
|---|---|
| ☒ | Antibodies |
| ☒ | Eukaryotic cell lines |
| ☒ | Palaeontology and archaeology |
| ☒ | Animals and other organisms |
| ☒ | Clinical data |
| ☒ | Dual use research of concern |
| ☒ | Plants |

## Methods

| n/a | Involved in the study |
|---|---|
| ☒ | ChIP-seq |
| ☒ | Flow cytometry |
| ☒ | MRI-based neuroimaging |

## Plants

| Seed stocks | *Report on the source of all seed stocks or other plant material used. If applicable, state the seed stock centre and catalogue number. If plant specimens were collected from the field, describe the collection location, date and sampling procedures.* |
|---|---|
| Novel plant genotypes | *Describe the methods by which all novel plant genotypes were produced. This includes those generated by transgenic approaches, gene editing, chemical/radiation-based mutagenesis and hybridization. For transgenic lines, describe the transformation method, the number of independent lines analyzed and the generation upon which experiments were performed. For gene-edited lines, describe the editor used, the endogenous sequence targeted for editing, the targeting guide RNA sequence (if applicable) and how the editor was applied.* |
| Authentication | *Describe any authentication procedures for each seed stock used or novel genotype generated. Describe any experiments used to assess the effect of a mutation and, where applicable, how potential secondary effects (e.g. second site T-DNA insertions, mosiacism, off-target gene editing) were examined.* |

