## [Peer Review File · Nature Genetics]

Genome-wide association study and polygenic risk prediction of hypothyroidism

Corresponding Author: Dr Søren Rand

Version 0:

Decision Letter:

3rd April 2024

Dear Dr. Rand,

Your Article "Genome-wide association study and polygenic risk prediction of hypothyroidism" has now been seen by two referees. You will see from their comments below that, while they find your work of interest, they have raised several relevant points. We are interested in the possibility of publishing your study in Nature Genetics, but we would like to consider your response to these points in the form of a revised manuscript before we make a final decision on publication.

To guide the scope of the revisions, the editors discuss the referee reports in detail within the team, including with the chief editor, with a view to identifying key priorities that should be addressed in revision, and sometimes overruling referee requests that are deemed beyond the scope of the current study. In this case, we ask that you put your work in context with the recent study by Sterenborg et al. (PMID: 38291025) and address all technical queries related to the association and polygenic risk score analyses, including details of phenotyping and how heterogeneity was accounted for in the analyses. We hope you will find this prioritized set of referee points to be useful when revising your study. Please do not hesitate to get in touch if you would like to discuss these issues further.

We therefore invite you to revise your manuscript taking into account all reviewer and editor comments. Please highlight all changes in the manuscript text file. At this stage, we will need you to upload a copy of the manuscript in MS Word .docx or similar editable format.

*2) If you have not done so already, please begin to revise your manuscript so that it conforms to our Article format instructions, available

http://www.nature.com/ng/authors/article_types/index.html here

*3) Include a revised version of any required Reporting Summary: <https://www.nature.com/documents/nr-reporting-summary.pdf>

Please be aware of our <https://www.nature.com/nature-research/editorial-policies/image-integrity> guidelines on digital image standards.

Link Redacted

We hope to receive your revised manuscript within 8-12 weeks. If you cannot send it within this time, please let us know.

Nature Genetics is committed to improving transparency in authorship. As part of our efforts in this direction, we are now requesting that all authors identified as 'corresponding author' on published papers create and link their Open Researcher and Contributor Identifier (ORCID) with their account on the Manuscript Tracking System (MTS), prior to acceptance. ORCID helps the scientific community achieve unambiguous attribution of all scholarly contributions. You can create and link your ORCID from the home page of the MTS by clicking on 'Modify my Springer Nature account'. For more information, please visit www.springernature.com/orcid.

Sincerely,
Kyle

Kyle Vogan, PhD
Senior Editor
Nature Genetics
<https://orcid.org/0000-0001-9565-9665>

Referee expertise:

Referee #1: Genetics, thyroid function

Referee #2: Genetics, thyroid function

Reviewers' Comments:

Reviewer #1:
Remarks to the Author:

Summary

The authors present a genetic association study of hypothyroidism, employing two strategies: a genome-wide case-control analysis of hypothyroidism, and testing of TSH-associated variants for association with hypothyroidism. Together, and thanks to the large sample size amassed, these strategies resulted in 156 novel genetic associations for hypothyroidism, of which 109 had not previously been associated with other thyroid traits such as TSH (though the latter figure may be reduced slightly further by the latest GWAS of TSH by Sterenborg et al., published just before submission of this manuscript). Testing in independent populations replicated 28 of the 156 at a Bonferroni-corrected threshold and 105 at a nominal P-value threshold. As expected, variant-to-gene mapping strategies highlight a central role for immunity and inflammation, and flag some potential targets for intervention.

The authors go on to develop and test a polygenic score for hypothyroidism. The area under the curve (AUC) for risk of hypothyroidism is similar to that previously demonstrated using a PRS trained on TSH measurements (Williams et al., 2023), but the current analysis demonstrates a substantially larger relative risk. The authors also show that a risk score incorporating genetics and anti-TPO antibody status predicts incident diagnoses of overt hypothyroidism amongst those with biochemically-defined subclinical hypothyroidism.

The most novel aspect of this analysis, besides the increase in yield of genetic associations, is the illustration of risk stratification based on a combination of "lifestyle" and genetic risk factors, showing a 10-year disease risk of 45% amongst females aged >60 with three key risk factors and in the top 10% of the PRS.

Comments to the authors

Major comments

1. The manuscript does not refer to the most recent GWAS of thyroid hormones by Sterenborg et al., which was published shortly before this article was submitted. Please review this study and take into account as appropriate when stating the number of unreported variants and placing other findings in context.

2. You have only applied genomic control to cohorts with marked inflation. Please provide justification, and consider adjusting the UK Biobank TSH analysis, which has an LDSC intercept of 1.11.
3. Methods: Please clarify how the prescription-based definition was operationalized in those cohorts where it was applied. For example, for UK Biobank, please state which drugs were extracted using which Read, BNF and dm+d codes, so that the approach can be fully understood and assessed by the reader and reproduced if they so wish.
4. Please ensure that you also describe and discuss the potential limitations of the phenotyping approach. For example, for UK Biobank, prescription data is available for less than half of the cohort, and secondary care data (ICD-10 codes) will only capture those cases where the diagnosis was coded during a hospital admission. In the portion of the sample where prescription data were available, how many were also captured by secondary care data? Does this suggest any degree of likely misclassification between cases and controls in the portion of the sample where prescription data were not available? Can you demonstrate that your strategy replicates previously published findings for hypothyroidism?
5. The phenotyping approach may also enrich the cases for hyperthyroidism, given that hyperthyroidism and related ICD-10 codes were excluded from the controls but not the cases, and that thyroxine prescription (or indeed diagnostic codes) used to identify cases could also encompass those taking thyroxine following radioactive iodine treatment or thyroid surgery. Please review the likely impact of this on the results and consider sensitivity analyses.
6. Supplementary Tables: For hypothyroidism, please give case and control numbers in each cohort. In a meta-analysis of five studies, I also think it would be reasonable (and informative) to include per-cohort GWAS results for the top signals from the meta-analysis in relevant supplementary tables.
7. Discussion: In the first paragraph, there are findings introduced that are not really evidenced in the results section. The reader has to interrogate the supplementary tables in some detail to identify (a) the number of associations with chemokine levels and (b) evidence for risk-mitigating variants correlated with lower levels of IFN-gamma, TNF-alpha and so on. Chemokines are a specific group of cytokines only referenced once in the results, in relation to associations with two SNPs in SH2B3 and PTPN11. In terms of risk-mitigating variants, those described in the results are two SNPs in TYK2 and STING1, whose protein associations are all with chemokines rather than the wider range of cytokines. I think you are referring to something more general in the discussion, but this is not currently clearly justified by the results section. Moreover, I have not found any evidence in the supplementary tables that any risk-reducing variants were associated with IFN-gamma levels specifically. Please review this section, remove any statements that are not justified by your findings, and ensure that key findings referred to in the discussion are also clearly described in the results.

Minor comments

1. Abstract: Please clarify that the 156 variants referenced are previously unreported for hypothyroidism, as around a third have previously been associated with other thyroid traits.
2. Introduction: The very start of the text refers to 5% of cases being undiagnosed. I think this represents a misunderstanding of the estimated population prevalence of undiagnosed cases of around 5%, described in the review by Chaker et al. It is also worth noting that this estimate, from a meta-analysis by Garmendia Madariaga et al, includes both subclinical and overt hypothyroidism, of which subclinical most likely represents the majority, so I question whether this is the most appropriate estimate to present as a headline even if described correctly. Please review the original study by Garmendia Madariaga et al and amend this statement.
3. Results: There is a minor error in the section "Polygenic risk score and hypothyroidism prediction" where you state that the "PRS significantly increased the Δ AUC by 8.3%" – this should presumably be AUC, not Δ AUC.
4. Methods: For the TSH analysis, please provide details of how the exclusions for thyroid drugs (ATC H03) and thyroid surgery were operationalized in each cohort (also see major comment on this).
5. Discussion: The third sentence of the discussion appears to suggest that associations with blood cell abundance and inflammatory markers highlight the role of hormonal feedback. I'd suggest separating out hormonal feedback from this sentence.
6. Discussion: Also in the first paragraph, I don't think it is correct to suggest that the leukocyte infiltration is a hallmark of autoimmunity rather than the inflammatory response in general. Please reword or remove.
7. Discussion: In the fifth paragraph, the hypothesized mechanism of immune response to the damaging effects of ultraviolet light appears somewhat speculative. You cite a review article from 2012 which discusses a likely effect of immune suppression by UV radiation on skin cancer, and also refers to epidemiological evidence that UV radiation also has a protective effect against autoimmunity, via separate mechanisms. Your argument goes beyond this in a number of ways, extending the link to cancers other than skin cancer, and suggesting a possible effect in the opposite direction of autoimmunity protecting against UV damage. Please provide additional evidence, or rephrase this claim much more cautiously.
8. Supplementary Tables: The text of the manuscript refers to 156 novel variants; the title of ST7 suggests 157 unreported

variants – please check and correct if needed.

Reviewer #2:

Remarks to the Author:

Rand et al. conducted a genome-wide association study and meta-analysis for hypothyroidism assessed via ICD codes, and for TSH and free thyroxine (fT4) levels. Several *in silico* follow-up analyses and lookups were conducted on the findings. A substantial emphasis was put on a polygenic risk score (PRS) relative to traditional risk factors in predicting incident disease, and progression from subclinical to overt hypothyroidism. The paper is well written and the analyses are generally sound. However, some aspects of the methods listed in detail below need to be specified. Although the results are interesting, I am not sure if the clinical application of the PRS will be as impactful as indicated by the authors. My major concerns are related to the question if the rather broad phenotype definition used in the current study disentangles the complexity of the hypothyroidism trait that will be required in precision medicine (i.e. using a PRS prediction).

The authors mention correctly the complex nature of hypothyroidism. However, ICD codes for autoimmune thyroiditis were not included in the case selection for the hypothyroidism GWAS, but instead the more general codes E03.8 ('Other specified hypothyroidism') and E03.9 ('Hypothyroidism, unspecified') in addition to thyroid medication intake. This trait heterogeneity applies similarly to the self-reported data of 23andMe. As a result, the observed contrary effects of smoking could be an indication that the phenotype definition applied in this project covers several subtypes of hypothyroidism.

In this study, smokers have a higher risk of developing hypothyroidism compared to a lower risk of getting autoimmune hypothyroidism reported in ref 24. This contradictory result could be related to the heterogeneity in the quite broad phenotype definition of hypothyroidism applied in the current study. This affects also the PRS and its results. Thus, I see limitations in the clinical application of the risk prediction in its current form.

Furthermore, the authors showed that the PRS can improve the prediction of hypothyroidism compared to established non-genetic risk factors. This predictive improvement of the PRS was substantially reduced if anti-TPO measurements are included in the model. How strong is the benefit of the PRS when thyroid hormone measurements like TSH and fT4 that are commonly used to diagnose hypothyroidism are additionally included in the model? This should be addressed.

The results of the recently published GWAS on thyroid function including TSH and free T4 (Sternberg et al., 2024. PMID: 38291025) should be considered for assessing known associations as this affects several of the newly reported hits (e.g. rs114322847 in TG).

Which criteria for replication were finally applied? It was written that 28 out of 152 (18%) variants replicated beyond the threshold for multiple testing. Consequently, the remaining variants were/should be considered as false positive associations?

The information on TSH and fT4 measurements are missing. Which assays were used? According to Table S1, TSH and fT4 values were also obtained from the UKB study. Were they measured in all individuals, only in hospital patients, etc.?

With respect to subclinical hypothyroidism, the authors state in the Discussion that the PRS could identify individuals who are most likely to benefit from early therapeutic interventions, and underscore the prevention strategies for hypothyroidism. How would these therapeutic interventions and prevention strategies look like keeping in mind the cumulative 10-year risk increment in the upper 10% PRS group of 50% (Figure 3)?

A PheWAS is predominantly assessing pleiotropic effects of the genetic variants. Thus, the statement that hypothyroidism may protect against various cancers is somewhat misleading and overstated with respect to cancer protection. Particularly, already a genetic score based on TSH values within the reference values revealed such associations with consistent effect directions (see Sternberg et al., 2024). Taking this into account, the genetic association with cancer could be driven by TSH associated variants.

How many genetic variants were included in the PRS obtained from the PRS-CS? Was this score also used for the PheWAS?

Minor issues:

The SumHer heritability estimates seem to be quite heterogeneous, e.g., the heritability of TSH in the ThyroidOmics meta-analysis is less than the TSH variance explained by the independent index SNPs in that study. Please comment on the reliability of the SumHer estimates.

The step 5 in the candidate gene prioritization is not quite clear. How were the posterior probabilities obtained from the FUSION TWAS p-values? Was subsequently a dedicated colocalization analysis conducted? If yes, please explain the rationale, and provide the corresponding methods. Furthermore, in Table S14 the outcome trait of the TWAS/colocalization should be stated.

The SNP effect size betas of TSH (section Correlation between thyroid hypothyroidism and thyroid hormones) are provided

in mU/L. Please state how this unit measures were obtained from the inverse-normal transformed TSH GWAS.

How was genome-wide significance defined? As $p < 1E-9$ as indicated in line 167, or as commonly used $5E-8$ in concordance with the reported findings?

Last sentence of the Introduction: As far as I understood, not a genetic correlation (i.e. using all variants) between hypothyroidism and common traits was assessed but an association of the PRS. Please check and correct the phrasing accordingly.

Table 1: Does EA/NEA really denote the minor and major allele or rather the effect/non-effect allele? In any case, the latter ones should be reported (which applies also for all Supplementary Tables).

Table S1: Please include the number of cases and controls of each study included in the hypothyroidism GWAS.

Table S20: The 23andMe Software is NA. Please provide information on the association software used.

Figure 5: the effect direction should be included in the plot.

Results line 169: Please provide the upper and lower bounds of the odd ratio IQR (not just 0.09).

Line 301 (Integrating genetic risk and lifestyle factors): I assume that "healthier" individual lifestyle characteristics were associated with a reduced risk of hypothyroidism. That should be added.

Version 1:

Decision Letter:

8th October 2024

Dear Søren,

Your revised Article "Genome-wide association study and polygenic risk prediction of hypothyroidism" has been seen by the original referees. You will see from their comments below that, while they find the study improved, they have raised a few additional points. We remain interested in the possibility of publishing your study in Nature Genetics, but we would like to consider your response to these points in the form of a revised manuscript before we make a final decision on publication.

As before, to guide the scope of the revisions, the editors discuss the referee reports in detail within the team, including with the chief editor, with a view to identifying key priorities that should be addressed in revision. In this case, we ask that you address the remaining points raised by each reviewer by clarifying details of the analyses and implementing appropriate revisions to the text and display items. We again hope that you will find the prioritized set of referee points to be useful when revising your study. Please do not hesitate to get in touch if you would like to discuss these issues further.

We therefore invite you to revise your manuscript taking into account all reviewer and editor comments. Please highlight all changes in the manuscript text file. At this stage, we will need you to upload a copy of the manuscript in MS Word .docx or similar editable format.

*2) If you have not done so already, please begin to revise your manuscript so that it conforms to our Article format instructions, available

[here](http://www.nature.com/ng/authors/article_types/index.html).

*3) Include a revised version of any required Reporting Summary: <https://www.nature.com/documents/nr-reporting-summary.pdf>

Please be aware of our [guidelines](https://www.nature.com/nature-research/editorial-policies/image-integrity) on digital image standards.

Link Redacted

We hope to receive your revised manuscript within 4-8 weeks. If you cannot send it within this time, please let us know.

Nature Genetics is committed to improving transparency in authorship. As part of our efforts in this direction, we are now requesting that all authors identified as 'corresponding author' on published papers create and link their Open Researcher and Contributor Identifier (ORCID) with their account on the Manuscript Tracking System (MTS), prior to acceptance. ORCID helps the scientific community achieve unambiguous attribution of all scholarly contributions. You can create and link your ORCID from the home page of the MTS by clicking on 'Modify my Springer Nature account'. For more information, please visit www.springernature.com/orcid.

Sincerely,
Kyle

Kyle Vogan, PhD
Senior Editor
Nature Genetics
<https://orcid.org/0000-0001-9565-9665>

Referee expertise:

Referee #1: Genetics, thyroid function

Referee #2: Genetics, thyroid function

Reviewers' Comments:

Reviewer #1 (Remarks to the Author):

My thanks to the authors for their thorough response and for the additional work undertaken to strengthen the analysis and manuscript. It is particularly interesting to see the improved power for discovery following the refinement of the phenotype.

In relation to the self-reported medication component of the new phenotype definition, it seems odd not to have included liothyronine (1140884512) when you have included other, less common codes for the same medication, but the numbers are small (142 individuals according to the UK Biobank showcase). For future analyses, it may also be worth including eltroxin (127 individuals) for completeness. These small numbers are unlikely to impact the results, and I don't think further revision is required.

In relation to the analysis of subclinical hypothyroidism, I am not sure how immortal time would occur, since the outcomes described in the new text are defined by EHR so are not limited to occurring after study enrolment, unless there are some other self-reported outcomes used for this analysis which are not described. If so, this should be clarified.

My other comments have been fully addressed and I have no further concerns.

Reviewer #2 (Remarks to the Author):

I congratulate the authors on this comprehensive rebuttal, and thank them for satisfactorily addressing all my questions, comments and concerns. It was a pleasure reading the manuscript. From my point of view, there are just a few minor issues that emerged during the revision and should be corrected:

- Suppl Methods UK Biobank Thyroid hormone analysis (p. 4): It should be added that the TSH and fT4 values were obtained from primary care data (which seems missing in the paragraph and is not obvious to the reader).

- Table S7: Please rename Graves' disease into Hyperthyroidism in the column header (at least Teumer et al. assessed subclinical hyperthyroidism and not explicitly Graves').
- Please cross check the assigned gene names – I assume it is labeled by the nearest gene, e.g., MICOS10 in Table S7 seems to be the commonly associated CAPZB locus.
- Please check the Locus numbers between Tables S3, S4, S13-S16, S18 as their numbers are not consistently assigned across these tables, e.g. for rs10917470.
- References 10 and 17 are duplicates.

Version 2:

Decision Letter:

Our ref: NG-A64728R1

14th January 2025

Dear Søren,

Thank you for submitting your revised manuscript "Genome-wide association study and polygenic risk prediction of hypothyroidism" (NG-A64728R1). In light of your responses to the points raised at the previous round of review, we will be happy in principle to publish your study in Nature Genetics as an Article pending final revisions to comply with our editorial and formatting guidelines.

We are now performing detailed checks on your paper, and we will send you a checklist detailing our editorial and formatting requirements soon. Please do not upload the final materials or make any revisions until you receive this additional information from us.

Thank you again for your interest in Nature Genetics. Please do not hesitate to contact me if you have any questions.

Sincerely,
Kyle

Kyle Vogan, PhD
Senior Editor
Nature Genetics
<https://orcid.org/0000-0001-9565-9665>

Manuscript ID: NG-A64728.R1

Manuscript title: Genome-wide association study and polygenic risk prediction of hypothyroidism

Point-by-point responses to comments from Reviewer #1.

The authors present a genetic association study of hypothyroidism, employing two strategies: a genome-wide case-control analysis of hypothyroidism, and testing of TSH-associated variants for association with hypothyroidism. Together, and thanks to the large sample size amassed, these strategies resulted in 156 novel genetic associations for hypothyroidism, of which 109 had not previously been associated with other thyroid traits such as TSH (though the latter figure may be reduced slightly further by the latest GWAS of TSH by Sterenborg et al., published just before submission of this manuscript). Testing in independent populations replicated 28 of the 156 at a Bonferroni-corrected threshold and 105 at a nominal P-value threshold. As expected, variant-to-gene mapping strategies highlight a central role for immunity and inflammation and flag some potential targets for intervention.

The authors go on to develop and test a polygenic score for hypothyroidism. The area under the curve (AUC) for risk of hypothyroidism is similar to that previously demonstrated using a PRS trained on TSH measurements (Williams et al., 2023), but the current analysis demonstrates a substantially larger relative risk. The authors also show that a risk score incorporating genetics and anti-TPO antibody status predicts incident diagnoses of overt hypothyroidism amongst those with biochemically-defined subclinical hypothyroidism.

The most novel aspect of this analysis, besides the increase in yield of genetic associations, is the illustration of risk stratification based on a combination of “lifestyle” and genetic risk factors, showing a 10-year disease risk of 45% amongst females aged >60 with three key risk factors and in the top 10% of the PRS.

Response: We thank the Reviewer for the thorough review of our work, and for providing clear and constructive comments. In our response highlighted in blue, we have taken the Reviewer’s comments into consideration, and modifications made directly to the manuscript as a result of the feedback are marked in red. We believe that these revisions overall strengthened the manuscript and the robustness of our findings.

Before addressing the Reviewer’s comments, we would like to inform that during the review process, our analysis sample size has increased. The Copenhagen Hospital Biobank has been updated with the addition of ~30,000 individuals. Also, we have received an updated version of the UKB data, adding 2 years of additional follow-up information (updated from September 2021 to March 2024). Third, FinnGen summary data was updated using the latest release (R10).

1. The manuscript does not refer to the most recent GWAS of thyroid hormones by Sterenborg et al., which was published shortly before this article was submitted. Please review this study and take into account as appropriate when stating the number of unreported variants and placing other findings in

context.

Response: Thank you for bringing this publication to our attention. We have taken the findings made by Sterenborg *et al.* into account and the number of ‘previously unreported loci’ has been updated accordingly.

Introduction, Page 4, Line 12: “The largest GWASs of thyroid stimulating hormone (TSH) and free thyroxine (fT4) have linked 159 and 65 genetic loci, respectively, to these traits.^{9–11} For hypothyroidism, 160 genetic loci have been associated to this disease.^{12–14}”

Results, Page 6, Line 8: “At genome-wide significance ($P < 5 \times 10^{-8}$), we identified 319 genome-wide significant loci, of which 150 were previously unreported (**Table 1, Fig. 1A, and Supplementary Table 2 and 3**) and 84 were not previously associated with other thyroid traits (**Supplementary Table 4**). Using a more stringent threshold of $P < 1 \times 10^{-9}$ we found 247 loci of which 86 were unreported.”

For hypothyroidism loci discovered in the hypothyroidism meta-analysis, **Supplementary Table 4, Columns W-AB** has been updated with the findings for TSH and fT4 made by Sterenborg *et al.*

Results, Page 6, Line 20: “In a meta-analysis of up to 191,449 individuals with fT4 measurements, we identified 61 fT4 genome-wide significant loci, of which 15 were previously unreported (**Supplementary Table 5**). In a meta-analysis of up to 482,873 individuals with TSH measurements, 297 TSH genome-wide significant loci were identified, 126 of which have not been previously reported (**Supplementary Table 6**).”

For fT4, **Supplementary Table 5, Column P-R** specifically lists on locus-level, whether the locus was reported in previous publications (including findings from Sterenborg *et al.*). For TSH, **Supplementary Table 6, Column P-R** also lists whether loci were reported in previous publications.

Results, Page 6, Line 31: “In total, we identified 350 nonoverlapping loci via hypothyroidism meta-analysis or through the TSH-driven approach (**Supplementary Table 7**), 179 of which have not been reported previously.”

For hypothyroidism loci discovered via hypothyroidism meta-analysis or the endophenotype-driven analysis, **Supplementary Table 7, Columns AF-AK** has been updated with the findings for TSH and fT4 made by Sterenborg *et al.*

2. You have only applied genomic control to cohorts with marked inflation. Please provide justification, and consider adjusting the UK Biobank TSH analysis, which has an LDSC intercept of 1.11.

Response: As suggested by the Reviewer, we have now applied genomic control to the UKB TSH analysis also. Please see below:

Methods, Page 18, Line 2: “We observed signs of inflation in FinnGen_{Hypo-GWAS} ($\lambda_{GC}=1.40$, LDSC-intercept = 1.21), CHB-CID/DBDS_{TSH-GWAS} ($\lambda_{GC} = 1.35$, LDSC-intercept = 1.19), and UKB_{TSH-GWAS} ($\lambda_{GC} = 1.24$, LDSC-intercept = 1.1) and accounted for potential bias by correcting the GWAS standard errors by the square root of the LDSC-intercept.⁴⁶ No additional genomic control was applied.”

3. Methods: *Please clarify how the prescription-based definition was operationalized in those cohorts where it was applied. For example, for UK Biobank, please state which drugs were extracted using which Read, BNF and dm+d codes, so that the approach can be fully understood and assessed by the reader and reproduced if they so wish.*

Response: We would like to point out some important changes before addressing this comment. First, prescription-based data was previously operationalized in the phenotypic definition of hypothyroidism in UKB but is no longer used in the updated phenotypic definition (see comment #4 for details). Further, we agree with the Reviewer that important data on drug codes is currently missing. In the updated version of our manuscript, we operationalize drug codes from self-reported data in the definition of hypothyroidism in UKB. Please see details in the changes made below:

For UK Biobank in **Supplementary Material, Page 3, Line 22:** “**Case and control definition.** We defined cases using the following ICD-10 codes: E03.8, E03.9 and E06.3 using electronic health records. Additional cases included individuals reporting hypothyroidism at baseline or using thyroid hormone medication. In the Data-Field 20003, the following codes indicated use of thyroid hormone medication; 1140874844 (tertroxin 20mcg tablet), 1140874852 (thyroxine sodium), 140884516 (thyroxine product), 1140910814 (sodium thyroxine), 1141191044 (levothyroxine sodium), 1140909904 (tri-iodothyronine product), 1140910518 (t3 – liothyronine), and 1140910520 (sodium liothyronine). Individuals diagnosed with ICD-10 E05[0-9] were excluded from the analysis. Controls were defined as free of the following ICD-10 codes: E01, E02, E03.0, E03.1, E03.2, E03.3, E03.4, E03.5, E04, E05, E06 and E07. All participants were of White European ancestry.”

Further, we also use drug codes in the GWA-studies of TSH and ft4. We have now detailed these in the **Manuscript** and in **Supplementary Tables**.

Methods, Page 17, Line 16: “Using CHB-CID/DBDS, UKB primary care data, and previously published data, we meta-analyzed GWA-studies for TSH and ft4. We used the first non-missing sample value that was within the reference range. The results of individual thyroid function tests were inverse normalized. Individuals who were either on thyroid drugs or had undergone thyroid surgery prior to the thyroid function tests were excluded. In the UK Biobank primary care data, thyroid hormone measurements were captured using Read2 and Read3 codes, while drug and operation codes were recorded using dm+d and OPCS-4 codes, respectively. In the CHB-DBDS, thyroid hormones were captured using NPU codes, drugs using ATC codes, and surgical procedures using procedure codes (**Supplementary Table 24**).”

Specifically, we have created a new table (**Supplementary Table 24**) that details the dm+d codes used for drugs and includes the OPCS-4 procedure codes, as well as Read2 and Read3 codes, used to define TSH and ft4 in the UK Biobank primary care data (please also see comment #14).

For the definition of the subclinical cohort, we have edited the **Manuscript** to clearly guide the reader on where to find necessary detailed information. We have also created a new table (**Supplementary Table 26**), which provides the full overview of the definition of the subclinical hypothyroidism cohort, including which dm+d, Read2, Read3, and ICD-10 codes were used to define this phenotype. These additions ensure that our approach can be fully understood, assessed, and reproduced by readers.

Methods, Page 21, Line 14: “**Disease progression in subclinical hypothyroidism patients.** In UKB primary care data, we defined individuals with SCH as having TSH levels greater than 4 mU/L and fT4 levels between 8 and 14.5 pmol/L using Read2 and Read3 codes. Only biochemical measurements available after the date of enrollment to UKB were used to avoid immortal time bias. Prior to the date for SCH, we excluded individuals with a history of thyroid cancer, hyperthyroidism, and hypothyroidism using ICD10, Read2, and Read3 codes and individuals taking thyroid-hormone substitution as indicated by dm+d codes (**Supplementary Table 26**). The PRS was categorized into three groups: $\leq 10^{\text{th}}$ percentile, representing low risk individuals, $> 10^{\text{th}}$ and $< 90^{\text{th}}$ percentiles representing the general population, and $\geq 90^{\text{th}}$ percentile representing high risk individuals. We used Cox regression models to compute HRs for risk of progression to overt hypothyroidism. Individual follow-up ended in the case of an event (defined by ICD10 E03.8/E03.9/E06.3 or Read2/Read3 codes indicative of autoimmune myxoedema/Hashimoto’s thyroiditis), death or end of follow-up, whichever occurred first. Models were adjusted for age, sex, and 4 PCs.”

4. Please ensure that you also describe and discuss the potential limitations of the phenotyping approach. For example, for UK Biobank, prescription data is available for less than half of the cohort, and secondary care data (ICD-10 codes) will only capture those cases where the diagnosis was coded during a hospital admission. In the portion of the sample where prescription data were available, how many were also captured by secondary care data? Does this suggest any degree of likely misclassification between cases and controls in the portion of the sample where prescription data were not available? Can you demonstrate that your strategy replicates previously published findings for hypothyroidism?

Response: We appreciate the Reviewer’s attention to potential misclassification using primary care data in UKB. Regarding the potential limitations of our phenotyping approach, we have carefully reconsidered the method and acknowledge that the UKB prescription data only partially covers the UKB study population, which could lead to potential misclassification. To mitigate these concerns, we have now prioritized self-reported medication data. This approach, while not without limitations, reduces the reliance on incomplete data. The list of thyroid hormone drugs operationalized to define a case has been added in **Supplementary Material**, and the **Methods** section covering phenotypic definitions for UKB has been revised to reflect the updated phenotypic definition.

Methods, Page 17, Line 3: “**Cohorts, association testing, and meta-analysis.** We used four cohorts for discovery (the Copenhagen Hospital Biobank Chronic Inflammatory Disease cohort and the Danish Blood Donor Study [CHB-CID/DBDS], UK Biobank [UKB], FinnGen Freeze 10, and 23andMe). The Estonian Biobank (EB) and deCODE genetics were used for validation. Cases were defined using International Classification of Disease (ICD)-10 codes E03.8/E03.9/E06.3, ICD-9 codes 244.8/244.9 or claimed prescription of thyroid hormone substitution therapy using Anatomical Therapeutic Chemical Classification (ATC) code H03A. In

UKB, in addition to electronic health registries, we included individuals self-reporting hypothyroidism or use of thyroid hormone substitution as cases. Individuals with hyperthyroidism (E05[0-9]) were excluded, and we otherwise used thyroid disease-free controls, excluding individuals with the following ICD-10 codes: E0[1-2], E03[0-5], and E0[4-7]. In 23andMe, cases were defined based on self-reported diagnoses of hypothyroidism, elevated thyroid-stimulating hormone, or taking levothyroxine. Controls were individuals who reported no other thyroid-related disorders.⁴⁴ Details on genotyping, imputation, and quality control are provided in the **Supplementary Material** and **Supplementary Table 23.**”

For UK Biobank in **Supplementary Material, Page 3, Line 22: “Case and control definition.** We defined cases using the following ICD-10 codes: E03.8/E03.9/ E06.3 using electronic health records. Additional cases included individuals reporting hypothyroidism at baseline or using thyroid hormone medication. In the Data-Field 20003, the following codes indicated use of thyroid hormone medication; 1140874844 (tertroxin 20mcg tablet), 1140874852 (thyroxine sodium), 140884516 (thyroxine product), 1140910814 (sodium thyroxine), 1141191044 (levothyroxine sodium), 1140909904 (tri-iodothyronine product), 1140910518 (t3 – liothyronine), and 1140910520 (sodium liothyronine). Individuals diagnosed with ICD-10 E05[0-9] were excluded from the analysis. Controls were defined as free of the following ICD-10 codes: E01, E02, E03.0, E03.1, E03.2, E03.3, E03.4, E03.5, E04, E05, E06 and E07. All participants were of White European ancestry.”

For the replication of previously published findings on hypothyroidism, we present below the genetic correlations between our results and those reported in previous GWASs. To demonstrate the robustness of our definition, we have selected three different studies of hypothyroidism which have used different phenotypic approaches: Dönertas HM (UKB, self-reported), Sakaue S (UKB, FinnGen, BBJ, using electronic health registries) and Mathieu S (UKB and FinnGen, using electronic health registries and levothyroxine purchases). We found near perfect correlation between beta-estimates of our lead variants and the beta-estimates in the highlighted studies.

Study	Cohorts	Case/control	Phenotypic definition
Dönertas HM, 2021 (PMID: 33959723)	UK Biobank	NA	Self-reported
Sakaue S, 2021 (PMID: 34594039)	Biobank Japan UK Biobank FinnGen	31,269/552,642 Europeans and East Asian	ICD10:DE03 in BBJ PheCode 244 in UKB Hypothyroidism in FinnGen
Mathieu S, 2022 (PMID: 36093044)	UK Biobank FinnGen	51,194/443,383	ICD9- or ICD10 E03 in UKB Hypothyroidism, levothyroxin purchases in FinnGen

Donertas HM (UKB, self-reported)

Sakaue S (UKB, FinnGen, BBJ, using electronic health registries)

Mathieu S (UKB and FinnGen, using electronic health registries and levothyroxine purchases)

5. The phenotyping approach may also enrich the cases for hyperthyroidism, given that hyperthyroidism and related ICD-10 codes were excluded from the controls but not the cases, and that thyroxine prescription (or indeed diagnostic codes) used to identify cases could also encompass those taking thyroxine following radioactive iodine treatment or thyroid surgery. Please review the likely impact of this on the results and consider sensitivity analyses.

Response: This is a highly important comment. First, to mitigate the risk of enrichment for hyperthyroidism, we redefined the phenotype, and specifically excluded individuals with ICD-10 E05[0-9], to ensure that cases of hypothyroidism due to secondary causes (e.g., hyperthyroidism) are excluded. To also accommodate Reviewer #2's comment #1 regarding omission of ICD codes for autoimmune thyroiditis, we also included ICD-10 E063 "Autoimmune thyroiditis/Hashimoto's thyroiditis" to the case definition, in cohorts where this information was available.

Methods, Page 17, Line 11: "Individuals with hyperthyroidism (E05[0-9]) were excluded, and we otherwise used thyroid disease-free controls, excluding individuals with ICD-10 codes: E0[1-2], E03[0-5], and E0[4-7]."

For Copenhagen Hospital Biobank in **Supplementary Material, Page 2, Line 12:** "**Case and control definition.** Cases were defined using International Classification of Disease (ICD)-10 codes: E03.8('Other specified hypothyroidism'), E03.9 ('Hypothyroidism, unspecified'), and E06.3 ('Autoimmune thyroiditis'/Hashimoto's thyroiditis') or using the following Anatomical Therapeutic Chemical (ATC) code: H03A. Individuals with ICD-10 E05[0-9] were excluded from the analysis. Controls were remaining individuals who were free of the following ICD-10 codes: E01, E02, E03.0, E03.1, E03.2, E03.3, E03.4, E03.5, E04, E05, E06 and E07."

For UK Biobank in **Supplementary Material, Page 3, Line 22:** "**Case and control definition.** We defined cases using the following ICD-10 codes: E03.8/E03.9/E06.3 using electronic health records. Additional cases included individuals reporting hypothyroidism at baseline or using thyroid hormone medication. In the Data-Field 20003, the following codes indicated use of thyroid hormone medication; 1140874844 (tertroxin 20mcg tablet), 1140874852 (thyroxine sodium), 140884516 (thyroxine product), 1140910814 (sodium thyroxine), 1141191044 (levothyroxine sodium), 1140909904 (tri-iodothyronine product), 1140910518 (t3 – liothyronine), and 1140910520 (sodium liothyronine). Individuals diagnosed with ICD-10 E05[0-9] were excluded from the analysis. Controls were defined as free of the following ICD-10 codes: E01, E02, E03.0, E03.1, E03.2, E03.3, E03.4, E03.5, E04, E05, E06 and E07. All participants were of White European ancestry."

For Estonian Biobank in **Supplementary Material, Page 5, Line 25:** "**Case and control definition.** Cases were defined using ICD-10 codes: E03.8/E03.9/E06.3 or using ATC code H03A. Individuals with ICD-10 E05[0-9] were excluded from the analysis. Controls were remaining individuals who were free of the following ICD-10 codes: E01, E02, E03.0, E03.1, E03.2, E03.3, E03.4, E03.5, E04, E05, E06 and E07. "

For deCODE genetics in **Supplementary Material, Page 7, Line 16:** "**Case and control definition.** Cases were defined using ICD-10 codes: E03.8, E03.9/E06.3, ICD-9 codes: 244[8-9], and ATC code H03A. Individuals with ICD-10 E05[0-9] were excluded from the analysis. Controls were thyroid-disease individuals free of the following ICD-10 codes: E01, E02, E03.0, E03.1, E03.2, E03.3, E03.4, E03.5, E04, E05, E06 and E07."

Collectively, we have reviewed the impact of adjusting the phenotypic definition, comparing results between the previous and current meta-analysis. The comparison indicates that the refined phenotype captures the genetic architecture more accurately:

- Improved genetic discovery:
 - In the hypothyroidism meta-analysis discovery increased from 301 to 319 loci.
 - Using the endophenotype-driven analysis discovery increased from 326 to 350 loci.
- Improved genetic discovery at more stringent thresholds:
 - Using a more stringent threshold of $P < 1 \times 10^{-9}$, we previously found 232 loci (73 unreported), which increased to 247 loci (86 unreported).
- Effect estimates from known Mendelian disease variants of hypothyroidism increased:
 - rs121908866 (stop-gain in *TSHR*) increased from OR 6.19 (95% CI 4.32 - 8.87) to OR 7.67 (95% CI 5.35 - 11.01).
- Higher genetic correlation with thyroid function traits:
 - TSH genetic correlation increased from 49% to 55%
 - fT4 genetic correlation decreased from -19% to -23%
- Improved prediction of hypothyroidism using a PRS derived from the update phenotype.
 - In UKB, the association on a linear basis increased from OR 1.95 per SD increase (95% CI 1.92 – 1.97, $P = 1.3 \times 10^{-2775}$) to OR 2.01 per SD increase (95% CI 1.99 – 2.03, $P = 2.3 \times 10^{-2790}$)
 - In GESUS, the association on a linear basis increased from OR 1.94 per SD increase (95% CI 1.79 - 2.10, $P = 7.7 \times 10^{-62}$) to OR 2.0 per SD increase (95%CI 1.85 – 2.17, $P = 1.61 \times 10^{-66}$)
 - In UKB when using the PRS for prediction of incident hypothyroidism, the Δ AUC increased from 6.5% (95%CI 6.0 – 6.9) to 7.2% (95% CI 6.7 – 7.6).

An overview is provided below in tabular form:

Key-highlights from previous meta-analysis results					
Genetic correlation TSH	Genetic correlation FT4	Hypothyroidism beta coefficients vs. TSH beta coefficients	Hypothyroidism beta coefficients vs. ft4 beta coefficients	PRS association to hypothyroidism UKB	PRS association to hypothyroidism GESUS
49% ($P = 6.01 \times 10^{-82}$)	-19% ($P = 0.01$)	55% ($P = 3.2 \times 10^{-26}$)	-10% ($P = 0.07$)	OR 1.95 per SD increase (95% CI 1.92 – 1.97, $P = 1.3 \times 10^{-2775}$)	OR 1.94 per SD increase (95% CI 1.79 - 2.10, $P = 7.7 \times 10^{-62}$)
Key-highlights from current meta-analysis					
Genetic correlation TSH	Genetic correlation FT4	Hypothyroidism beta coefficients vs. TSH beta coefficients	Hypothyroidism beta coefficients vs. ft4 beta coefficients	PRS association to hypothyroidism UKB	PRS association to hypothyroidism GESUS
55% ($P = 3.5 \times 10^{-122}$)	-23% ($P = 0.003$)	58% ($P = 3.65 \times 10^{-33}$)	-16% ($P = 0.005$)	OR 2.01 per SD increase (95% CI 1.99 – 2.03, $P = 2.3 \times 10^{-2790}$)	OR 2.0 per SD increase (95%CI 1.85 – 2.17, $P = 1.61 \times 10^{-66}$)

6. Supplementary Tables: For hypothyroidism, please give case and control numbers in each cohort. In a meta-analysis of five studies, I also think it would be reasonable (and informative) to include per-cohort GWAS results for the top signals from the meta-analysis in relevant supplementary tables.

Response: We have included case/control and N for quantitative traits to Supplementary Table 1, Column J. A per-cohort GWAS results has been added in an additional table, Supplementary Table 2.

7. Discussion: In the first paragraph, there are findings introduced that are not really evidenced in the results section. The reader has to interrogate the supplementary tables in some detail to identify (a) the number of associations with chemokine levels and (b) evidence for risk-mitigating variants correlated with lower levels of IFN-gamma, TNF-alpha and so on. Chemokines are a specific group of cytokines only referenced once in the results, in relation to associations with two SNPs in SH2B3 and PTPN11. In terms of risk-mitigating variants, those described in the results are two SNPs in TYK2 and STING1, whose protein associations are all with chemokines rather than the wider range of cytokines. I think you are referring to something more general in the discussion, but this is not currently clearly justified by the results section. Moreover, I have not found any evidence in the supplementary tables that any risk-reducing variants were associated with IFN-gamma levels specifically. Please review this section, remove any statements that are not justified by your findings, and ensure that key findings referred to in the discussion are also clearly described in the results.

Response: We appreciate the Reviewer's constructive feedback on the Discussion and have carefully revised the Results and Discussion sections in response (please also see comment #15 and #16). We have now specified and clearly described the results and discussed these findings in greater detail. Specifically, we have emphasized the associations of the risk-mitigating variants with inflammatory mediators and put their significance in the context of autoimmune diseases related to hypothyroidism. Additionally, we have ensured that the Discussion aligns with our results. We would like to clarify that the interferon-gamma (IFN- γ) is included in our supplementary results, but it is listed under its protein symbol "IFNG". We apologize for any confusion this may have caused.

Results, Page 8, Line 17: "Next, we interrogated variants associated with lower hypothyroidism risk in genes with known roles in immune system function. We highlight two missense variants, rs149007883 in *NFKBIZ* (p.Gly102Ala, OR 0.83) and rs34536443 (OR 0.87, p.Pro1104Ala) in *TYK2*, and two intron variants rs13181561 (OR 0.96) in *STING1* and rs113473633 (OR 0.90) in *NKFB1*. These variants associated with lower levels of a panel of inflammatory mediators (**Supplementary Fig. 2**), including IFN- γ , CXCL10 and CXCL9, which make up key pathogenic pathways involved in autoimmune diseases related to hypothyroidism.^{19,20}"

Supplementary Fig. 2, Supplementary Material, Page 11: "Heatmap for four hypothyroidism risk-mitigating variants with known regulatory roles in the immune system. A colored box marks a nominal significant association ($P < 0.05$) and an asterisk marks a significant association beyond threshold for multiple testing ($P < 1.41 \times 10^{-6}$ [$0.05/350$ variants \times 101 traits]). A blue colored box indicates lower levels of an inflammatory marker, whereas a red box indicates higher levels."

Discussion, Page 13, Line 13: “We highlight associations with inflammatory markers which may provide insight into inflammatory pathogenic mechanisms.²⁷ We emphasize four risk-mitigating variants (missense; rs149007883 in *NFKBIZ*, rs34536443 in *TYK2*, and intron; rs13181561 in *STING1*, and rs113473633 in *NKFB1*) in genes encoding critical regulators of immune system function.^{28–31} These variants were associated with lower levels of inflammatory mediators, including IFN- γ , CXCL10 and CXCL9, that make up a crucial pathway in the activation and recruitment immune cells.^{19,20} This is proposed to be a central pathogenic pathway in many autoimmune diseases, including vitiligo, psoriasis, and psoriatic arthritis, which are closely linked to hypothyroidism. This aligns with the increased expression of IFN- γ in serum and thyroid tissue of patients with autoimmune hypothyroidism, proposed to mediate thyroid hormone deficiency via lymphocyte infiltration and the exposure of thyrocytes to proinflammatory cytokines.^{4,32–34} Further investigation into key signalling pathways, such as those highlighted, may be critical for understanding the mechanisms underlying disease pathogenesis.³⁴”

Minor comments

11. Abstract: Please clarify that the 156 variants referenced are previously unreported for hypothyroidism, as around a third have previously been associated with other thyroid traits.

Response: This is indeed ambiguous, and we have edited the abstract to clarify that the 179 (the updated number) variants referenced are previously unreported for hypothyroidism.

Abstract, Page 3, Line 5: “We identified 350 loci associated with hypothyroidism, including 179 that have not been previously reported for hypothyroidism.”

12. Introduction: The very start of the text refers to 5% of cases being undiagnosed. I think this represents a misunderstanding of the estimated population prevalence of undiagnosed cases of around 5%, described in the review by Chaker et al. It is also worth noting that this estimate, from a meta-analysis by Garmendia Madariaga et al, includes both subclinical and overt hypothyroidism, of which subclinical most likely represents the majority, so I question whether this is the most appropriate estimate to present as a headline even if described correctly. Please review the original study by Garmendia Madariaga et al and amend this statement.

Response: We have revised the Introduction and Discussion to reflect the findings of Garmendia Madariaga et al., who distinguish between subclinical and overt hypothyroidism:

Introduction, Page 4, Line 3: “Primary hypothyroidism is a common and insidious metabolic disease. It is characterized by subtle and nonspecific symptoms, which can lead to delayed diagnosis, resulting in an underdiagnosed case burden estimated at up to 0.5%.”

Discussion, Page 14, Line 9: “An estimated 0.5% of individuals with undiagnosed hypothyroidism may reflect individuals who do not seek medical attention for gradually developing nonspecific symptoms.”

13. Results: *There is a minor error in the section “Polygenic risk score and hypothyroidism prediction” where you state that the “PRS significantly increased the Δ AUC by 8.3%” – this should presumably be AUC, not Δ AUC.*

Response: We agree with the Reviewer that these results would benefit from clearer wording. We have made the following changes:

Results, Page 10, Line 10: “We next evaluated the predictive ability of the PRS relative to established risk factors.⁶ Relative to a model with age, sex, and principal components (PCs), the PRS gave the largest change in area under the curve (Δ AUC) of 7.2% (95% CI 6.7 – 7.6), which exceeded the impact of all other risk factors (**Fig. 2B**). Integrating all non-genetic risk factors into a model resulted in a Δ AUC of 0.5% (95% CI 0.4 – 0.7), and a model including all risk factors (including the PRS) resulted in a Δ AUC of 7.8% (95% CI 7.3 – 8.2; AUC 0.70).”

14. Methods: *For the TSH analysis, please provide details of how the exclusions for thyroid drugs (ATC H03) and thyroid surgery were operationalized in each cohort (also see major comment on this).*

Response: We agree that this information is highly relevant and as also addressed in comment #3, we have described the phenotypic definition in greater detail in the **Manuscript** and added all details to a new table (**Supplementary Table 24**). This table details Read2, and -3 codes used to define TSH and fT4 in the UK Biobank primary care data, and also details which dm+d and OPCS-4 codes that were used for exclusion. For CHB-CID/DBDS, this table also includes the NPU-codes that were used to define TSH and fT4 alongside with ATC and surgical procedure codes that were used for exclusion.

Methods, Page 17, Line 18: “We used the first non-missing sample value that was within the reference range. The results of individual thyroid function tests were inverse normalized. Individuals who were either on thyroid drugs or had undergone thyroid surgery prior to the thyroid function tests were excluded. In the UK Biobank primary care data, thyroid hormone measurements were captured using Read2 and Read3 codes, while drug and operation codes were recorded using dm+d and OPCS-4 codes, respectively. In the CHB-DBDS, thyroid hormones were captured using NPU codes, drugs using ATC codes, and surgical procedures using procedure codes (**Supplementary Table 24**).”

15. Discussion: *The third sentence of the discussion appears to suggest that associations with blood cell abundance and inflammatory markers highlight the role of hormonal feedback. I’d suggest separating out hormonal feedback from this sentence.*

Response: We have made several changes to the Discussion. Please see changes below and in the following comments:

Discussion, Page 13, Line 3: “In this study, we present the largest genetic evaluation of thyroid hormone deficiency to date, through GWAS meta-analyses of hypothyroidism and thyroid hormones. Our findings confirm and extend the understanding of the polygenic and complex nature of hypothyroidism, linking three-hundred-fifty genetic loci to this disease. By linking genetic loci to immune-related cells and circulating inflammation markers, we characterized potential mediators of disease. Using gene-prioritization methods, we identified putatively casual genes with known roles in autoimmunity which aligns with the main etiology in iodine-sufficient areas of the world.²⁶ We showed that using a hypothyroidism PRS could potentially improve the diagnostic accuracy in thyroid hormone deficiency, a condition fraught with diagnostic challenges.”

16. Discussion: *Also in the first paragraph, I don't think it is correct to suggest that the leukocyte infiltration is a hallmark of autoimmunity rather than the inflammatory response in general. Please reword or remove.*

Response: The discussion section has been thoroughly revised specifically after addressing comment #7 and #15. We put emphasis on variants, that associated with inflammatory mediators, which make up pathological pathways involved in other autoimmune diseases and their significance in the context of autoimmune diseases related to hypothyroidism and directly to hypothyroidism.

Discussion, Page 13, Line 13: “We highlight associations with inflammatory markers which may provide insight into inflammatory pathogenic mechanisms.²⁷ We emphasize four risk-mitigating variants (missense; rs149007883 in *NFKBIZ*, rs34536443 in *TYK2*, and intron; rs13181561 in *STING1*, and rs113473633 in *NKFB1*) in genes encoding critical regulators of immune system function.^{28–31} These variants were associated with lower levels of inflammatory mediators, including IFN- γ , CXCL10 and CXCL9, that make up a crucial pathway in the activation and recruitment immune cells.^{19,20} This is proposed to be a central pathogenic pathway in many autoimmune diseases, including vitiligo, psoriasis, and psoriatic arthritis, which are closely linked to hypothyroidism. This aligns with the increased expression of IFN- γ in serum and thyroid tissue of patients with autoimmune hypothyroidism, proposed to mediate thyroid hormone deficiency via lymphocyte infiltration and the exposure of thyrocytes to proinflammatory cytokines.^{4,32–34} Further investigation into key signalling pathways, such as those highlighted, may be critical for understanding the mechanisms underlying disease pathogenesis.³⁴”

17. Discussion: *In the fifth paragraph, the hypothesized mechanism of immune response to the damaging effects of ultraviolet light appears somewhat speculative. You cite a review article from 2012 which discusses a likely effect of immune suppression by UV radiation on skin cancer, and also refers to epidemiological evidence that UV radiation also has a protective effect against autoimmunity, via separate mechanisms. Your argument goes beyond this in a number of ways, extending the link to cancers other than skin cancer, and suggesting a possible effect in the opposite direction of autoimmunity*

protecting against UV damage. Please provide additional evidence, or rephrase this claim much more cautiously.

Response: We agree with the Reviewer that this argument is overstated, and that additional and stronger evidence is required to support such a claim. This point was also noted by Reviewer #2 in comment #8. We have carefully revised the discussion to ensure it accurately represents our findings. Additionally, we have cross-referenced the discussion with relevant epidemiological studies and the study by Sterenborg et al., as suggested in comment #1.

Discussion, Page 15, Line 10: “Furthermore, we found significant associations between genetically predicted higher hypothyroidism risk and lower risk of breast, prostate, and skin cancer, supporting the findings reported by several observational studies.^{42,43} The association between the PRS and breast cancer aligns with that of a recent GWAS of thyroid function.¹¹ Interestingly, we found no association between hypothyroidism risk and thyroid cancer, despite previous GWASs showing an association between higher TSH levels and lower risk of thyroid cancer.^{11,16} Whether the observed associations with specific cancers reflect shared pathways, in which an augmented immunosurveillance on one hand leads to both disease but on the other hand mitigates the risk of specific cancers will require additional investigation.”

18. Supplementary Tables: The text of the manuscript refers to 156 novel variants; the title of ST7 suggests 157 unreported variants – please check and correct if needed.

Response: We have thoroughly revised the **Supplementary Tables** to ensure the correctness of naming, legends, and contents. Because the updated version of the manuscript consists of additional data, and the fact that we have taken newly published data into consideration, this number has also been changed.

Point-by-point responses to comments from Reviewer #2.

Rand et al. conducted a genome-wide association study and meta-analysis for hypothyroidism assessed via ICD codes, and for TSH and free thyroxine (fT4) levels. Several in silico follow-up analyses and lookups were conducted on the findings. A substantial emphasis was put on a polygenic risk score (PRS) relative to traditional risk factors in predicting incident disease, and progression from subclinical to overt hypothyroidism. The paper is well written and the analyses are generally sound. However, some aspects of the methods listed in detail below need to be specified. Although the results are interesting, I am not sure if the clinical application of the PRS will be as impactful as indicated by the authors. My major concerns are related to the question if the rather broad phenotype definition used in the current study disentangles the complexity of the hypothyroidism trait that will be required in precision medicine (i.e. using a PRS prediction).

Response: We appreciate the positive and constructive feedback and thorough review of our work, particularly regarding heterogeneity in the analysis. We have carefully considered all suggestions and implemented substantial changes throughout the manuscript. The adjustments made to the phenotypic definition have strengthened our findings, and the Reviewers' comments have significantly improved the presentation of our research. Our detailed responses are provided in blue below, and the revised text in the manuscript is shown in red.

Before addressing the Reviewer's comments, we would like to inform that during the review process, our analysis sample size has increased. The Copenhagen Hospital Biobank has been updated with the addition of ~30,000 individuals. Also, we have received an updated version of the UKB data, adding 2 years of additional follow-up information (updated from September 2021 to March 2024). Third, FinnGen summary data was updated using the latest release (R10).

1. The authors mention correctly the complex nature of hypothyroidism. However, ICD codes for autoimmune thyroiditis were not included in the case selection for the hypothyroidism GWAS, but instead the more general codes E03.8 ('Other specified hypothyroidism') and E03.9 ('Hypothyroidism, unspecified') in addition to thyroid medication intake. This trait heterogeneity applies similarly to the self-reported data of 23andMe. As a result, the observed contrary effects of smoking could be an indication that the phenotype definition applied in this project covers several subtypes of hypothyroidism.

Response: We appreciate that the Reviewer is raising these important points. In light of this comment and those of Reviewer #1 regarding phenotypic definition, we have made significant adjustments to our analyses. As the Reviewer points out, trait heterogeneity was inherent when including summary data from sources such as FinnGen (40% of cases) and 23andMe (15% of cases), where the ability to refine the phenotypic definition was limited. However, we believe the current changes have reduced such heterogeneity, which is reflected by the improvement in effect estimates for *bona fide* hypothyroidism loci (e.g., rs121908866 stop-gain in *TSHR*) and polygenic risk prediction. Specifically, we omitted individuals with hyperthyroidism (ICD-10 E05[0-9]) from our analyses. Additionally, as suggested, we have included individuals diagnosed with autoimmune thyroiditis (ICD-10 E063). Please see the changes made below:

Methods, Page 17, Line 3: “Cohorts, association testing, and meta-analysis. We used four cohorts for discovery (the Copenhagen Hospital Biobank Chronic Inflammatory Disease cohort and the Danish Blood Donor Study [CHB-CID/DBDS], UK Biobank [UKB], FinnGen Freeze 10, and 23andMe). The Estonian Biobank (EB) and deCODE genetics were used for validation. Cases were defined using International Classification of Disease (ICD)-10 codes E03.8/E03.9/E06.3, ICD-9 codes 244.8/244.9 or claimed prescription of thyroid hormone substitution therapy using Anatomical Therapeutic Chemical Classification (ATC) code H03A. In UKB, in addition to electronic health registries, we included individuals self-reporting hypothyroidism or use of thyroid hormone substitution as cases. Individuals with hyperthyroidism (E05[0-9]) were excluded, and we otherwise used thyroid disease-free controls, excluding individuals with the following ICD-10 codes: E0[1-2], E03[0-5], and E0[4-7]. ”

For Copenhagen Hospital Biobank in **Supplementary Material, Page 2, Line 12: “Case and control definition.** Cases were defined using International Classification of Disease (ICD)-10 codes: E03.8(‘Other specified hypothyroidism’), E03.9 (‘Hypothyroidism, unspecified’), and E06.3 (‘Autoimmune thyroiditis’/‘Hashimoto’s thyroiditis’) or using the following Anatomical Therapeutic Chemical (ATC) code: H03A. Individuals with ICD-10 E05[0-9] were excluded from the analysis. Controls were remaining individuals who were free of the following ICD-10 codes: E01, E02, E03.0, E03.1, E03.2, E03.3, E03.4, E03.5, E04, E05, E06 and E07.”

For UK Biobank in **Supplementary Material, Page 3, Line 22: “Case and control definition.** We defined cases using the following ICD-10 codes: E03.8/E03.9/E06.3 using electronic health records. Additional cases included individuals reporting hypothyroidism at baseline or using thyroid hormone medication. In the Data-Field 20003, the following codes indicated use of thyroid hormone medication; 1140874844 (tertroxin 20mcg tablet), 1140874852 (thyroxine sodium), 140884516 (thyroxine product), 1140910814 (sodium thyroxine), 1141191044 (levothyroxine sodium), 1140909904 (tri-iodothyronine product), 1140910518 (t3 – liothyronine), and 1140910520 (sodium liothyronine). Individuals diagnosed with ICD-10 E05[0-9] were excluded from the analysis. Controls were defined as free of the following ICD-10 codes: E01, E02, E03.0, E03.1, E03.2, E03.3, E03.4, E03.5, E04, E05, E06 and E07. All participants were of White European ancestry.”

For Estonian Biobank in **Supplementary Material, Page 5, Line 25: “Case and control definition.** Cases were defined using ICD-10 codes: E03.8/E03.9/E06.3 or using ATC code H03A. Individuals with ICD-10 E05[0-9] were excluded from the analysis. Controls were remaining individuals who were free of the following ICD-10 codes: E01, E02, E03.0, E03.1, E03.2, E03.3, E03.4, E03.5, E04, E05, E06 and E07. ”

For deCODE genetics in **Supplementary Material, Page 7, Line 16: “Case and control definition.** Cases were defined using ICD-10 codes: E03.8, E03.9/E06.3, ICD-9 codes: 244[8-9], and ATC code H03A. Individuals with ICD-10 E05[0-9] were excluded from the analysis Controls were thyroid-disease individuals free of the following ICD-10 codes: E01, E02, E03.0, E03.1, E03.2, E03.3, E03.4, E03.5, E04, E05, E06 and E07.”

As the Reviewer knows, autoimmune thyroiditis is the most common cause of hypothyroidism in iodine-sufficient areas (including all cohorts in this study). However, in cohorts where electronic health records were available to us, only a small proportion of participants had a diagnosis of autoimmune thyroiditis, as

ICD-10 code E063 captures a disproportionately small number of cases relative to the population prevalence. For example, in the Copenhagen Hospital Biobank, the prevalence of hypothyroidism using E063 was 0.7%, and in the UK Biobank, it was only 0.1%. This underutilization of the diagnosis code necessitated the use of other ICD-10 codes to reflect disease prevalence more accurately. While this approach may encompass subtypes, it primarily represents autoimmune hypothyroidism. We would like to bring to the Reviewers' attention that several previously published GWAS on hypothyroidism have employed a nearly identical approach to define the phenotype:

Study	Cohorts	Case/control	Phenotypic definition
Denny JC, 2011 (PMID: 21981779)	eMERGE	1,317/5,503	Thyroid replacement medication > three months and at least one International Classification of Disease, 9th edition code for hypothyroidism (all causes). Secondary causes of hypothyroidism and other thyroid conditions excluded
Eriksson N, 2012 (PMID: 22493691)	23andMe	3,736/35,546	Self-reported
Pickrell JK, 2016 (PMID: 27182965)	23andMe	17,558/117,083	Self-reported
Kichaev G, 2018 (PMID: 30595370)	UK Biobank	NA	NA
Saevarsdottir S, 2020 (PMID: 32581359)	UK Biobank deCODE	30,234/725,172	Autoimmune thyroid disease combining Graves' disease (E05.9) or Hashimoto's thyroiditis (E06.3) or hypothyroidism (E03.9) or (ATC code H03AA01). Secondary causes of hypothyroidism excluded
Dönertas HM, 2021 (PMID: 33959723)	UK Biobank	NA	Self-reported
Sakaue S, 2021 (PMID: 34594039)	Biobank Japan UK Biobank FinnGen	31,269/552,642 Europeans and East Asian	ICD10:DE03 in BBJ PheCode 244 in UKB Hypothyroidism in FinnGen
Mathieu S, 2022 (PMID: 36093044)	UK Biobank FinnGen	51,194/443,383	ICD9- or ICD10 E03 in UKB Hypothyroidism, levothyroxin purchases in FinnGen

2. In this study, smokers have a higher risk of developing hypothyroidism compared to a lower risk of getting autoimmune hypothyroidism reported in ref 24. This contradictory result could be related to the heterogeneity in the quite broad phenotype definition of hypothyroidism applied in the current study. This affects also the PRS and its results. Thus, I see limitations in the clinical application of the risk prediction in its current form.

Response: We appreciate the Reviewer addressing concerns in regard to how heterogeneity might influence the association between hypothyroidism risk and smoking. First, to mitigate any bias attributed to including individuals with hyperthyroidism, we redefined the PRS validation cohorts (UKB and GESUS) and specifically excluded individuals with ICD-10 E05[0-9], to ensure that cases of hypothyroidism due to secondary causes (e.g., hyperthyroidism) were excluded. Moreover, as mentioned under #1, our phenotype definition is similar to that of previous GWA-studies. We have made the following changes:

Methods, Page 20, Line 20: “We validated the PRS in UKB, where individuals with ICD-10 E05[0-9] were removed to mitigate enrichment for participants with hyperthyroidism amongst hypothyroidism cases.”

Methods, Page 21, Line 1: “The PRS was also evaluated in the Danish General Suburban Population Study (GESUS). This was a population-based cohort study in which 21,205 adults were recruited between 2010 and 2013. At baseline, participants underwent physical examination, completed a questionnaire, and had blood samples drawn. Individuals with ICD10 E05[0-9] were excluded from the analysis.”

With regards to the association directionality between smoking and hypothyroidism, we revisited our analyses after the updated phenotypic definition. We did not observe any material changes to direction of effect or association between lifestyle factors and hypothyroidism risk.

Results, Page 11, Line 11: “We found that healthy lifestyle characteristics were associated with a reduced risk of hypothyroidism. As expected, individuals without obesity had lower risk (HR 0.71, 95% CI 0.68-0.74) compared to obese individuals.²⁴ Contrary to previous findings, we found that nonsmokers had a lower risk (HR 0.79, 95% CI 0.75 – 0.84) than did current smokers.²⁵ Overall, adherence to a healthy lifestyle corresponded to an HR of 0.83 (95% CI 0.79 – 0.87), while an unhealthy lifestyle corresponded to an HR of 1.26 (95% CI 1.16 – 1.35; **Fig. 4A** and **Supplementary Table 21**). Finally, we explored the interplay between the PRS and lifestyle factors to identify individuals at extreme disease risk (**Fig. 4B** and **4C**). The 10-year risk was greater for women, with the highest risk observed along the PRS axis. Risk increased with accumulating risk factors and higher polygenic risk, where the highest 10-year risk (50%) was observed for women above the age of 60 years, with a PRS in the >90th percentile of the distribution, who were obese, smokers, and did not exercise regularly.”

In regard to the Reviewers concern related to the clinical application of the PRS, we appreciate the chance to elaborate further on this matter. After the updated phenotypic definition, we observed stronger effect estimates for hypothyroidism. Please see results below. We have edited the **Manuscript** throughout with these improved results.

Association to hypothyroidism on a linear basis (OR per SD increase)			
Previous PRS		Current PRS	
In UKB	In GESUS	In UKB	In GESUS
OR 1.95 (95% CI 1.92 – 1.97, P = 1.3×10^{-2775})	OR 1.94 (95% CI 1.79 - 2.10, P = 7.7×10^{-62})	OR 2.01 (95% CI 1.99 – 2.03, P = 2.3×10^{-2790})	OR 2.0 (95%CI 1.85 – 2.17, P = 1.61×10^{-66})

Prediction of incident hypothyroidism in UKB	
Previous PRS	Current PRS
ΔAUC of 6.5% (95% CI 6.0 – 6.9)	ΔAUC of 7.2% (95% CI 6.7 – 7.6)

In this study, we showed that the PRS together with easily accessible lifestyle factors can identify 50% of those who will develop disease during follow-up. Moreover, by utilizing data on anti-TPO together with thyroid hormones in the GESUS cohort, which were measured for all individuals, and importantly *not* on clinical indication, allowed us to conduct a head-to-head comparison with usual clinical care. The ability of the PRS to improve risk prediction in a context where both autoantibodies *and* thyroid hormones are taken into account, we believe demonstrates the potential of PRSs for disease prediction.

Results, Page 10, Line 16: “In the GESUS cohort, we identified 5,452 individuals with TSH, fT4, and anti-TPO measurements, that were free of hypothyroidism at baseline. Of these, 431 were anti-TPO positive (> 100 U/mL). A model including age, sex, and PCs yielded an AUC of 0.634 (95% CI 0.589 - 0.679). A model including thyroid hormones and anti-TPO increased AUC further to 0.849 (95% CI 0.810 - 0.889). By adding the PRS to the latter model, we improved risk prediction, significantly increasing the AUC to 0.859 (95% CI 0.821 – 0.897, *P* for difference = 0.03; **Supplementary Table 20**).“

Another main finding was that the PRS was able to predict the clinical course of subclinical hypothyroidism. In the current version of the manuscript, the PRS had its high 10-year risk prediction, also after adding a low-risk group, which now enables risk stratification of progression in low, medium and high-risk individuals (which is further specified in comment #7):

Results, Page 11, Line 1: “Compared to individuals with intermediate polygenic risk (>10th to 90th percentiles), individuals with high polygenic risk (>90th percentile) had an HR 1.43 (95% CI 1.37 – 1.61) for progressing to overt hypothyroidism, while low risk individuals (<10th percentile) had an HR of 0.76 (95% CI 0.65 – 0.88). On the absolute scale, this risk translated to a 10.2% higher 10-year conversion rate for high risk individuals (39.3%, 95% CI 35.9 - 42.7%) and a 6.6% lower 10-year conversion rate for low risk individuals (22.5%, 95% CI 19.6 - 25.4%) compared to those in the intermediate risk group (29.1%, 95% CI 28.0 - 30.3%, **Fig.3**).“

The combination of self-reported data, use of biobanks with limited freedom to define phenotypes, and underused ICD-10 billing for autoimmune hypothyroidism, may lead to some degree of heterogeneity in our analysis, which we now consider in the discussion of limitations:

Discussion, Page 15, Line 20: “This study has several limitations. First, the analysis was limited to individuals of European ancestry, which restricts the generalizability of our findings to other ancestries. Second, we relied on data from cohorts, in which the phenotype definition was based on self-reported diagnoses, for example 23andMe, or summary statistics with predefined phenotypes, limiting our ability to further refine the phenotype definitions. This may have introduced some degree of heterogeneity.“

3. Furthermore, the authors showed that the PRS can improve the prediction of hypothyroidism compared to established non-genetic risk factors. This predictive improvement of the PRS was substantially reduced if anti-TPO measurements are included in the model. How strong is the benefit of the PRS when thyroid hormone measurements like TSH and ft4 that are commonly used to diagnose hypothyroidism are additionally included in the model? This should be addressed.

Response: This comment brings significant clinical aspects to light. Using the Danish GESUS cohort, we were able to address this by adding 4 levels of risk factors stepwise to a prediction model consisting of age, sex and 4 PCs. These results were added to a new table, **Supplementary Table 20**. We found that adding the PRS to a model with which included age, sex, thyroid hormones, and anti-TPO significantly improved risk prediction. See changes made below:

Results, Page 10, Line 16: “Anti-TPO is a strong predictor of autoimmune hypothyroidism.²³ In the GESUS cohort, we identified 5,452 individuals with TSH, ft4, and anti-TPO measurements, that were free of hypothyroidism at baseline. Of these, 431 were anti-TPO positive (> 100 U/mL). A model including age, sex, and PCs yielded an AUC of 0.634 (95% CI 0.589 - 0.679). A model including thyroid hormones and anti-TPO increased AUC further to 0.849 (95% CI 0.810 - 0.889). By adding the PRS to the latter model, we improved risk prediction, significantly increasing the AUC to 0.859 (95% CI 0.821 – 0.897, *P* for difference = 0.03; **Supplementary Table 20**).”

In **Supplementary Table 20**, we describe in greater detail variables included, and the AUC and Δ AUC for each prediction model.

4. The results of the recently published GWAS on thyroid function including TSH and free T4 (Sterenborg et al., 2024. PMID: 38291025) should be considered for assessing known associations as this affects several of the newly reported hits (e.g. rs114322847 in TG).

Response: We have now taken the results reported by Sterenborg *et al* into account and updated the number of “novel” risk loci accordingly throughout the main text and **Supplementary Tables**.

Results, Page 6, Line 8: “At genome-wide significance ($P < 5 \times 10^{-8}$), we identified 319 genome-wide significant loci, of which 150 were previously unreported (**Table 1**, **Fig. 1A**, and **Supplementary Table 2 and 3**) and 84 were not previously associated with other thyroid traits (**Supplementary Table 4**). Using a more stringent threshold of $P < 1 \times 10^{-9}$ we found 247 loci of which 86 were unreported.”

For hypothyroidism loci discovered in the hypothyroidism meta-analysis, **Supplementary Table 4, Columns W-AB** has been updated with the findings for TSH and ft4 made by Sterenborg *et al*.

Results, Page 6, Line 20: “In a meta-analysis of up to 191,449 individuals with ft4 measurements, we identified 61 ft4 genome-wide significant loci, of which 15 were previously unreported (**Supplementary Table 5**). In a meta-analysis of up to 482,873 individuals with TSH measurements, 297 TSH genome-wide significant loci were identified, 126 of which have not been previously reported (**Supplementary Table 6**).”

For ft4, **Supplementary Table 5, Column P-R** specifically lists on locus-level, whether the locus was reported in previous publications (including findings from Sterenborg *et al.*). For TSH, **Supplementary Table 6, Column P-R** also lists whether loci were reported in previous publications.

Results, Page 6, Line 31: “In total, we identified 350 nonoverlapping loci via hypothyroidism meta-analysis or through the TSH-driven approach (**Supplementary Table 7**), 179 of which have not been reported previously.”

For hypothyroidism loci discovered via hypothyroidism meta-analysis or the endophenotype-driven analysis, **Supplementary Table 7, Columns AF-AK** has been updated with the findings for TSH and ft4 made by Sterenborg *et al.*

5. Which criteria for replication were finally applied? It was written that 28 out of 152 (18%) variants replicated beyond the threshold for multiple testing. Consequently, the remaining variants were/should be considered as false positive associations?

Response: We thank the Reviewer for the question and for the opportunity to clarify our replication criteria. We performed power calculations for variants with odds ratios of 1.03, 1.05, 1.08, and 1.10, considering a minor allele frequency (MAF) range from 0.01 to 0.3. With a significance threshold of $\alpha = 0.00028$ ($0.05/179$ unreported variants), our replication analysis is well-powered to detect variants with $MAF > 0.1$ and $OR > 1.05$, as it is demonstrated in **Supplementary Fig. 1** and **Supplementary Table 8** to demonstrate the power calculations. The sample size for the discovery meta-analysis was 113,393 cases and 1,065,268 controls, while the replication cohort comprised of 34,835 cases and 492,149 controls. Given the much smaller sample size of the replication cohorts compared to the discovery datasets, we did not expect all of the SNPs to be strongly associated. Rather, if they were genuine associations, we would expect the effect estimates to be highly concordant, with some of the smaller ORs being individually nonsignificant. The concordance between the discovery cohort and our replication cohorts log ORs was high (correlation 85.3% $P = 6.5E-51$), indicating that our discovery GWAS identified genuine signals. To add an additional dimension of validation, we also cross-referenced the novel variants with GWAS data on TSH and ft4. These results have been added to **Supplementary Table 9**.

Results, Page 7, Line 4: “**Replication.** We replicated unreported variants in Estonian Biobank (EB) and deCODE genetics, which included 34,835 cases and 492,149 controls. Of the 179 novel loci reported here, 176 (98%) were available for replication. Thirty-five out of 176 (19%) variants replicated beyond the threshold for multiple testing ($P < 2.79 \times 10^{-4}$ [$0.05/179$]). A total of 110 out of 176 (63%) were nominally significant ($P < 0.05$), and all but one had concordant direction of effect. Finally, 54/176 (31%) had $P \geq 0.05$ but showed concordant direction of effect. There was a high concordance between effect estimates in the discovery and replication cohorts for the 179 risk variants ($r^2 = 0.85$, $P = 6.54 \times 10^{-51}$). Given the large sample size differences between discovery and replication, we did not expect to be able to replicate all novel loci at the threshold for multiple testing. Power calculations indicated that our replication analysis had limited power to detect variants with an OR of 1.03, which corresponds to the effect range of replication variants (**Supplementary Fig. 1** and **Supplementary Table 8**). We also cross-referenced variants that replicated at nominal significance ($P < 0.05$) with genome-wide associations to TSH and ft4. Of the 75

variants that replicated at nominal significance ($P < 0.05$), 32 were previously genome-wide significant in either TSH or fT4 studies. Of the 54 variants that did not replicate ($P \geq 0.05$), but had concordant direction of effect, 23 were genome-wide associated with either TSH or fT4 (**Supplementary Table 9**)."

Supplementary Fig. 1, Supplementary Material, Page 10: "Power analysis for replication of unreported genetic variants in deCODE/Estonian Biobank. Power analysis for detection of novel genetic hypothyroidism variants at various minor allele frequencies (MAF) and odds ratios (OR), using a significance threshold of $\alpha = 0.00028$ (0.05/179 unreported variants). Replication was performed in deCODE/Estonian Biobank, comprising 34,835 cases and 492,149 controls. Power calculations were conducted for ORs of 1.03 (blue line), 1.05 (green line), 1.08 (red line), and 1.10 (purple line) across a MAF range of 0.01 to 0.3. The horizontal red dashed line indicates the 80% power threshold."

6. The information on TSH and fT4 measurements are missing. Which assays were used? According to Table S1, TSH and fT4 values were also obtained from the UKB study. Were they measured in all individuals, only in hospital patients, etc.?

Response: In the UKB, we extracted data on TSH and fT4 from primary care data. Unfortunately, details on assays are not available. Samples included in CHB-CID/DBDS were primarily from hospital laboratories and originate from in or out-patient hospital visits and referrals from primary care. TSH and fT4 samples were measured on assays from Cobas (Roche Diagnostics), Alinity (Abbot Laboratories), Centaur XPT (Siemens Advia), and Dimension Vista 1500 (Siemens). All laboratories in Denmark are accredited by DANAK (following European standard ISO 15189), which ensures that all instruments follow the same procedures for quality control. This has been added to **Supplementary Material**:

Supplementary Material, Page 2, Line 22: "TSH and fT4 samples were measured on assays from Cobas (Roche Diagnostics), Alinity (Abbot Laboratories), Centaur XPT (Siemens Advia), and Dimension Vista 1500

(Siemens) from different laboratories in Denmark. All laboratories in Denmark are accredited by DANAK (following European standard ISO 15189), which ensures that all instruments follow the same procedures for quality control.”

7. With respect to subclinical hypothyroidism, the authors state in the Discussion that the PRS could identify individuals who are most likely to benefit from early therapeutic interventions, and underscore the prevention strategies for hypothyroidism. How would these therapeutic interventions and prevention strategies look like keeping in mind the cumulative 10-year risk increment in the upper 10% PRS group of 50% (Figure 3)?

Response: We agree that the statement concerning individual-level benefits is rather speculative. We have conducted additional analyses and revised the results and the discussion hereof. Specifically, we added individuals with low risk of progression (<10th percentile of the PRS). With these results in mind, we agree that rather than discussing prevention and intervention for individuals in different risk categories, our results demonstrate that polygenic risk may assist clinicians in selecting patient who are more or less likely to progress from one disease state to another. If and when genotyping become standard of care, the clinical consequence could be genotype-guided biochemical assessment, rather than the current practice, which is largely based on unspecific symptoms.

Results, Page 10, Line 29: “We identified 8,114 individuals from UKB primary care data with biochemically defined SCH and investigated whether the PRS could identify individuals who are more or less likely to progress to overt disease. Compared to individuals with intermediate polygenic risk (>10th to 90th percentiles), individuals with high polygenic risk (>90th percentile) had an HR 1.43 (95% CI 1.37 – 1.61) for progressing to overt hypothyroidism, while low risk individuals (<10th percentile) had an HR of 0.76 (95% CI 0.65 – 0.88). On the absolute scale, this risk translated to a 10.2% higher 10-year conversion rate for high risk individuals (39.3%, 95% CI 35.9 - 42.7%) and a 6.6% lower 10-year conversion rate for low risk individuals (22.5%, 95% CI 19.6 - 25.4%) compared to those in the intermediate risk group (29.1%, 95% CI 28.0 - 30.3%, Fig.3). “

Figures, Manuscript, Page 40: “**Fig. 3: Progression from subclinical hypothyroidism to overt disease.** Ten-year cumulative incidence of disease progression from subclinical hypothyroidism to overt hypothyroidism in 8,114 primary care patients from the UK Biobank. The green line represents the incidence in individuals with low polygenic risk (<10th percentile), the yellow line represents individuals with intermediate polygenic risk (10th - 90th percentile), and red represents individuals with high polygenic risk (>90th percentile). We used the Aalen-Johansen estimator, which accounts for the competing risk of death. Additionally, hazard ratios (HRs) were calculated using Cox regression models adjusted for age, sex, and 4 principal components.”

Discussion, Page 14, Line 23: “However, the clinical course of SCH to overt disease is unpredictable and relies on vague and nonspecific symptoms. We demonstrated that the PRS could identify individuals at high and low risk of progression from SCH to overt disease. If genotyping becomes a standard of care, polygenic risk scores may guide clinicians in selecting patients who are more or less likely to progress from one disease state to another. Consequently, the clinical approach could shift to a genotype-guided biochemical assessment, rather than relying solely on nonspecific symptoms to guide testing.”

8. A PheWAS is predominantly assessing pleiotropic effects of the genetic variants. Thus, the statement that hypothyroidism may protect against various cancers is somewhat misleading and overstated with respect to cancer protection. Particularly, already a genetic score based on TSH values within the reference values revealed such associations with consistent effect directions (see Sterenborg et al., 2024). Taking this into account, the genetic association with cancer could be driven by TSH associated variants.

Response: We agree that the discussion of these results was misrepresentative of our findings and that the conclusions made were notional. Also accommodating Reviewer #1’s comment #17, we have changed the discussion of these results, considering the findings by Sterenborg et al., and cross-referencing relevant epidemiological studies.

Discussion, Page 15, Line 10: “Furthermore, we found significant associations between genetically predicted higher hypothyroidism risk and lower risk of breast, prostate, and skin cancer, supporting the findings reported by several observational studies.^{42,43} The association between the PRS and breast cancer aligns with that of a recent GWAS of thyroid function.¹¹ Interestingly, we found no association between hypothyroidism risk and thyroid cancer, despite previous GWASs showing an association between higher TSH levels and lower risk of thyroid cancer.^{11,16} Whether the observed associations with specific cancers reflect shared pathways, in which an augmented immunosurveillance on one hand leads to both disease but on the other hand mitigates the risk of specific cancers will require additional investigation.”

9. How many genetic variants were included in the PRS obtained from the PRS-CS? Was this score also used for the PheWAS?

Response: In total, 1,107,248 variants were included in the PRS obtained from PRS-CS. This score was also utilized for the PheWAS analysis and has been specified in the results.

Results, Page 10, Line 1: “**Polygenic risk score and hypothyroidism prediction.** We derived a PRS of 1,107,248 variants from a meta-analysis of CHB-CID/DBDS, deCODE genetics, EB, FinnGen, and 23andMe, including more than 116,000 hypothyroidism cases.”

Results, Page 11, Line 24: “**Associations with cancer and cardiometabolic phenotypes.** We investigated the relationship between the hypothyroidism PRS and 50 phenotypes, including common malignancies and cardiometabolic traits in UKB.”

Minor issues:

10. The SumHer heritability estimates seem to be quite heterogeneous, e.g., the heritability of TSH in the ThyroidOmics meta-analysis is less than the TSH variance explained by the independent index SNPs in that study. Please comment on the reliability of the SumHer estimates.

Response: We appreciate the Reviewer's concern regarding the reliability of the heritability estimates used in our study. To complement and to be able to compare these estimates, we also calculated SNP-heritability on the liability scale using LDSC. While the heritability estimates were comparable for fT4 and TSH, as the Reviewer alluded to, heritability based on SumHer for hypothyroidism was slightly higher (see the figure below for an overview). As the authors behind SumHer (Speed *et al.*, Nature Genetics, 2018) describe in the paper, the most likely explanation for discrepancies between SumHer and LDSC is attributed to the underlying heritability models. In the paper, they show that LDK model for estimating heritability is better at capturing true heritability, whereas LDSC tends to underestimate it. Differences in modeling approaches explains why the SumHer estimates appear higher and underscores the importance of considering multiple methods for a comprehensive assessment. As a result, the LDSC heritability estimates, and standard errors have been added in **Supplementary Table 1, Column H & I.**

Comparison of Heritability Estimates (SumHer vs LDSC) by Trait

11. The step 5 in the candidate gene prioritization is not quite clear. How were the posterior probabilities obtained from the FUSION TWAS p-values? Was subsequently a dedicated colocalization analysis conducted? If yes, please explain the rationale, and provide the corresponding methods. Furthermore, in Table S14 the outcome trait of the TWAS/colocalization should be stated.

Response: We concur with the Reviewer's suggestion to provide further specification regarding the methods covering TWAS with colocalization. We have now documented the use of this method in greater detail, as shown below:

Methods, Page 19, Line 28: “Transcriptome-wide association study (TWAS) with colocalization: Using FUSION⁵⁴, we performed TWASs using hypothyroidism summary data, to investigate the relationship between the risk loci and effects on gene expression (eQTL) in GTEx v8 datasets on hypothalamus, pituitary gland, thyroid, spleen, whole blood, and pancreas. We used the internal colocalization function to detect shared causal variants between hypothyroidism risk and gene expression, which employs COLOC.⁵⁵ We only considered eQTLs associated with hypothyroidism at $P < 2.97 \times 10^{-6}$ (0.05/16,841 genes tested). Finally, we report the posterior probability of colocalized associations (PP4), which show evidence of a shared causal variant found in both GWAS and functional associations. If a hypothyroidism risk locus (sentinel variant \pm 1Mb) harbored a gene with $PP4 > 0.75$, we considered this a mediator of hypothyroidism and evidence of gene mapping.”

12. The SNP effect size betas of TSH (section Correlation between thyroid hypothyroidism and thyroid hormones) are provided in mU/L. Please state how this unit measures were obtained from the inverse-normal transformed TSH GWAS.

Response: We thank the Reviewer for catching this typo. We have replaced our typo with standard deviations instead.

Results, Page 7, Line 25: “For example, the missense variant rs78534766 in *ADCY7*, and the *FLT3* variant rs76428106 associated with large effects on hypothyroidism (OR 1.4 and 1.37, respectively) but had a comparably small effect on TSH levels (0.04 and 0.08 SD, respectively). Similarly, the variants rs2016105 in *ELK3* ($\beta = 0.17$ SD) and rs1479567 in *PDE8B* ($\beta = 0.16$ SD) had large effects on TSH but associated with a modest increase in disease risk (OR \sim 1.1; **Fig. 1B** and **Supplementary Table 10**).”

13. How was genome-wide significance defined? As $p < 1E-9$ as indicated in line 167, or as commonly used $5E-8$ in concordance with the reported findings?

Response: We consider genome-wide significance as $P < 5 \times 10^{-8}$ (as stated in the method section), but also included a secondary “threshold” to highlight loci with even stronger associations, while still acknowledging those meeting conventional threshold. We revised the results for improved clarity on this point:

Methods, Page 18, Line 8: “We considered a risk locus novel if no genome-wide significant association ($P < 5 \times 10^{-8}$) with hypothyroidism or the use of thyroid hormone replacement therapy had been reported previously.”

Results, Page 6, Line 8: “At genome-wide significance ($P < 5 \times 10^{-8}$), we identified 319 loci, of which 150 were previously unreported (**Table 1, Fig. 1A, and Supplementary Table 2 and 3**) and 84 were not previously associated with other thyroid traits (**Supplementary Table 4**). Using a more stringent threshold of $P < 1 \times 10^{-9}$ we found 247 loci of which 86 were unreported.”

14. Last sentence of the Introduction: As far as I understood, not a genetic correlation (i.e. using all variants) between hypothyroidism and common traits was assessed but an association of the PRS. Please check and correct the phrasing accordingly.

Response: We thank the Reviewer for pointing this out. We have now specified this in the introduction accordingly.

Introduction, Page 5, Line 8: “Finally, we explored the association between the hypothyroidism PRS and common malignancies, cardiometabolic, and neuropsychiatric traits.”

15. Table 1: Does EA/NEA really denote the minor and major allele or rather the effect/non-effect allele? In any case, the latter ones should be reported (which applies also for all Supplementary Tables).

Response: We thank the Reviewer for this. We have now replaced the minor/major alleles with “effect alleles” and “non-effect alleles” throughout the **Manuscript, Table 1, and Supplementary Tables**.

16. Table S1: Please include the number of cases and controls of each study included in the hypothyroidism GWAS.

Response: We thank the Reviewer for the suggestion and have addressed this under Reviewer #1’s comment #6, where we have included the number of cases and controls, as well as the sample size for quantitative traits in **Supplementary Table 1, Column J**.

17. Table S20: The 23andMe Software is NA. Please provide information on the association software used.

Response: Thank you for bringing this to our attention, this has now been edited in the corresponding **Supplementary Table 24**.

Supplementary Table 24, J-column, row 7: “Internal software packages.”

18. Figure 5: the effect direction should be included in the plot.

Response: Including the effect direction in **Fig. 5** would indeed improve its interpretation. The effect direction has now been added to the figure legend.

Figures, Manuscript, Page 42:

19. Results line 169: Please provide the upper and lower bounds of the odd ratio IQR (not just 0.09).

Response: We have added the 1st and 3rd quartile ranges to provide a clearer understanding of the variability:

Results, Page 6, Line 12: “Most lead single nucleotide polymorphisms (SNPs) had modest effect sizes (median odds ratio [OR] 1.03, interquartile range 0.96 – 1.05).”

20. Line 301 (Integrating genetic risk and lifestyle factors): I assume that “healthier” individual lifestyle characteristics were associated with a reduced risk of hypothyroidism. That should be added.

Response: We appreciate the Reviewer's observation, which has been added to the manuscript:

Results, Page 11, Line 11: “We found that healthy lifestyle characteristics were associated with a reduced risk of hypothyroidism.”

Manuscript ID: NG-A64728.R2

Manuscript title: Genome-wide association study and polygenic risk prediction of hypothyroidism

Point-by-point responses to comments from Reviewer #1.

My thanks to the authors for their thorough response and for the additional work undertaken to strengthen the analysis and manuscript. It is particularly interesting to see the improved power for discovery following the refinement of the phenotype.

In relation to the self-reported medication component of the new phenotype definition, it seems odd not to have included liothyronine (1140884512) when you have included other, less common codes for the same medication, but the numbers are small (142 individuals according to the UK Biobank showcase). For future analyses, it may also be worth including eltroxin (127 individuals) for completeness. These small numbers are unlikely to impact the results and I don't think further revision is required.

Response: We appreciate the Reviewer's positive feedback on our adjustments, and we once again thank the Reviewer for the thorough review of our work. As previously done, we highlight our response in blue and our modifications made directly to the manuscript as a result of the feedback are marked in red.

We thank the Reviewer for this observation and agree that including liothyronine and eltroxin would complete the medication component in the phenotype definition. These medications were unintentionally omitted; however we agree that these small numbers are unlikely to affect the results substantially.

In relation to the analysis of subclinical hypothyroidism, I am not sure how immortal time would occur, since the outcomes described in the new text are defined by EHR so are not limited to occurring after study enrolment, unless there are some other self-reported outcomes used for this analysis which are not described. If so, this should be clarified.

Response: In the SCH-to-hypothyroidism analysis, we only included thyroid hormone measurements that were analyzed after study enrollment. Similarly, hypothyroidism events that occurred after the onset of biochemically defined SCH were used. This is important in the competing risk setting, since including SCH defined before study enrollment would be an "immortal" period until study enrollment. Individuals with hypothyroidism before SCH diagnosis were of course also excluded. We have clarified this further:

Methods, Page 21, Line 16: "We only considered biochemical measurements available after the date of enrollment to UKB to avoid immortal time bias. Prior to the date for SCH, we excluded individuals with a history of thyroid cancer, hyperthyroidism, and hypothyroidism using ICD10, Read2, and Read3 codes and individuals taking thyroid-hormone substitution as indicated by dm+d codes (**Supplementary Table 26**). The PRS was categorized into three groups: \leq 10th percentile, representing low risk individuals, $>$ 10th and $<$ 90th percentiles representing the general population, and \geq 90th percentile representing high risk individuals. We used Cox regression models to compute HRs for risk of progression to overt hypothyroidism. Individual follow-up ended in the case of an event sampled following the date of SCH from

electronic health records (defined by ICD10 E03.8/E03.9/E06.3 or Read2/Read3 codes indicative of autoimmune myxoedema/Hashimoto's thyroiditis), death or end of follow-up, whichever occurred first. Models were adjusted for age, sex, and 4 PCs. Absolute risks were calculated using the Aalen-Johansen estimator, which takes the competing risk of death into account."

My other comments have been fully addressed and I have no further concerns.

Point-by-point responses to comments from Reviewer #2.

I congratulate the authors on this comprehensive rebuttal, and thank them for satisfactorily addressing all my questions, comments and concerns. It was a pleasure reading the manuscript. From my point of view, there are just a few minor issues that emerged during the revision and should be corrected:

Response: We appreciate the positive feedback and thorough review of our work, and thank the Reviewer for contributing to the improvement of the manuscript. Our response to the last minor edits and additions are detailed in blue below, and the revised text in the manuscript is shown in red.

Suppl Methods UK Biobank Thyroid hormone analysis (p. 4): *It should be added that the TSH and fT4 values were obtained from primary care data (which seems missing in the paragraph and is not obvious to the reader).*

Response: We now explicitly detail that we use the primary care dataset within UK Biobank covering ~ 245,000 participants whose data were available for non-Covid research.

Supplementary Table 1, Legend: *“The UK Biobank primary care data covered ~ 245,000 participants whose data were available for non-Covid research.”*

Supplementary Material, Page 4, Line 3: *“In UKB primary care data covering ~ 245,000 participants, we used the first non-missing sample value that was within the reference range.”*

Methods, Page 17, Line 17: *“Using CHB-CID/DBDS, UKB primary care data (the subset allowed for non-Covid research) and previously published data, we meta-analyzed GWA-studies for TSH and fT4.”*

Methods, Page 21, Line 14: *“In UKB primary care data (N ~ 245,000), we defined individuals with SCH as having TSH levels greater than 4 mU/L and fT4 levels between 8 and 14.5 pmol/L using Read2 and Read3 codes.”*

And in **Supplementary Table 24, A-D column, row 2:** *“UK Biobank primary care data”*

In Supplementary Table 24, Legend: *“Definition of thyroid stimulating hormone and free thyroxine in UK Biobank (UKB) primary care data (N ~ 245,000), and Copenhagen Hospital Biobank (CHB).”*

In Supplementary Table 26, Legend: *“Using the UK Biobank primary care data, we defined individuals with subclinical hypothyroidism using Read2 and Read3 codes for thyroid hormones.”*

Table S7: *Please rename Graves’ disease into Hyperthyroidism in the column header (at least Teumer et al. assessed subclinical hypothyroidism and not explicitly Graves’).*

Response: We have corrected this and renamed as requested.

Supplementary Table 7, W-Y column, row 3: “Hyperthyroidism”

Please cross check the assigned gene names – I assume it is labeled by the nearest gene. E.g., MICOS10 in Table S7 seems to be the commonly associated CAPZB locus.

Response: We have revised this, and all variants now map to the nearest gene, which has also been added to the columns of Supplementary Tables throughout.

Please check the Locus numbers between Tables S3, S4, S13-S16, S18 as their numbers are not consistently assigned across these tables, e.g. for rs10917470.

Response: After reviewing this point, we concluded that reporting Locus numbers before and after endophenotype analysis would cause more confusion than clarity. As a result, we omitted Locus numbers from all tables and instead report the top variant.

References 10 and 17 are duplicates.

Response: We thank the Reviewer for spotting this! We have removed the duplicated reference and corrected the citation in the text.